# Symmetry-Driven Discovery of Dynamical Variables in Molecular Simulations

**Jeet Mohapatra** [1]   **Nima Dehmamy** [2]   **Csaba Both** [3]   **Subhro Das** [2]   **Tommi Jaakkola** [1]

## Abstract

We introduce a novel approach for discovering effective degrees of freedom (DOF) in molecular dynamics simulations by mapping the DOF to approximate symmetries of the energy landscape. Unlike most existing methods, we do not require trajectory data but instead rely on knowledge of the forcefield (energy function) around the initial state. We present a scalable symmetry loss function compatible with existing force-field frameworks and a Hessian-based method efficient for smaller systems. Our approach enables systematic exploration of conformational space by connecting structural dynamics to energy landscape symmetries. We apply our method to two systems, Alanine dipeptide and Chignolin, recovering their known important conformations. Our approach can prove useful for efficient exploration in molecular simulations with potential applications in protein folding and drug discovery.

## 1. Introduction

Molecular dynamics (MD) is an essential tool for a wide range of applications, including drug discovery (Jorgensen, 2004; Hollingsworth & Dror, 2018), protein folding (Shaw et al., 2010; Lindorff-Larsen et al., 2011), and understanding the physics of biological systems at the molecular level (Karplus & Kuriyan, 2005; McCammon et al., 1977). The configuration space of molecules is very high-dimensional, with each atom contributing three degrees of freedom (xyz coordinates). However, the physically relevant conformations of molecules typically occupy a much lower-dimensional subspace. This space corresponds to regions of low free energy, which can be thought of as a negative log-likelihood of the system's state. The

most likely conformations correspond to the minima of this energy landscape. Sampling from this low-energy subspace is crucial for understanding the function and stability of biomolecules, but it presents significant computational challenges due to the vast number of possible configurations and the presence of energy barriers separating different conformational states. Existing methods for sampling these low-energy conformations include enhanced sampling techniques such as metadynamics (Laio & Parrinello, 2002), umbrella sampling (Torrie & Valleau, 1977), and replica exchange molecular dynamics (REMD) (Sugita & Okamoto, 1999), which aim to overcome energy barriers and improve exploration of the conformational space. However, these methods often require careful tuning and can be computationally expensive. More recently, machine learning has shown great success in enhanced transition path sampling (Holdijk et al., 2024; Sipka et al., 2023). Our work is complementary to these efforts, providing a data-free way to discover reaction coordinates or collective variables.

In this work, we present a novel approach for discovering degrees of freedom (DOF) that effectively move the system along the low-energy manifold, enabling more efficient exploration of relevant conformations in the molecular landscape. Our key observation is that low-energy DOF can be related to approximate symmetries of the energy function. Recent works in machine learning have made progress in discovering symmetries in data (Benton et al., 2020; Dehmamy et al., 2021; Yang et al., 2023b;a). Our setting is different in that we do not have data, but an energy function. The problem of finding symmetries of the energy function overlaps with the task addressed in LieGG (Moskalev et al., 2022), which discovers infinitesimal symmetry generators, i.e. Lie algebra elements, for a given loss function. In our case, we are interested in transformations which change the energy below a certain threshold, meaning approxiamte symmetries of the energy. For example, we don't want change in energy to be so high that it would break chemical bonds.

We find approximate symmetries by considering small transformations of the original DOF and deriving conditions for the near-invariance of the energy. This process yields a symmetry loss, which we then minimize. In the case of small molecules, we show that the problem can be

[1]Massachusetts Institute of Technology, Cambridge, MA, USA [2]MIT-IBM Watson AI Lab, Cambridge, MA, USA [3]NorthEastern University, Boston, MA, USA. Correspondence to: Jeet Mohapatra <jeetmo@mit.edu>.

*Proceedings of the 42$^{nd}$ International Conference on Machine Learning*, Vancouver, Canada. PMLR 267, 2025. Copyright 2025 by the author(s).

formulated as finding symmetries of the Hessian. Aside from the optimization approach, we also provide an analytic solution based on degenerate Hessian eigenspaces.

Our method connects the structural dynamics of molecules to the approximate symmetries of the energy landscape. This enables a systematic exploration of conformational space that is both computationally efficient and physically insightful. We use our method on Alanine dipeptide and Chignolin. We find that these effective DOFs can be used to efficiently explore the phase space of the molecule. In some cases, the DOF significantly overlap with chemical features such as dihedral angles. Hence, these DOFs facilitate targeted exploration and sampling of biologically relevant conformations. We show that this method of sampling can significantly reduce the computational burden associated with high-dimensional MD simulations. It also enhances the ability to explore areas of the conformational landscape that are difficult to access through conventional methods.

Our contributions can be summarized as follows.

1. **DOF as Symmetries:** We formulate identifying effective DOF (EDOF) as an optimization problem aiming to discover approximate symmetries of the energy.

2. **Symmetry discovery using the Hessian:** We derive the relation between symmetry generators and the Hessian near critical points. We also provide a method for constructing symmetries from the spectrum of the Hessian.

3. **Extraction of internal coordinates for molecules:** We show that our discovered DOF can overlap with well-known dihedral angles, but also include additional DOF.

4. **Exploring conformations of Alanine dipeptide and Chignolin:** We show that our DOF can be used to sample diverse conformations, particularly shallow local minima, which are difficult to sample using existing methods.

## 2. Related Work

Exploring the conformation space of molecular systems poses significant challenges due to their numerous degrees of freedom and the highly nonconvex energy landscape. Traditional molecular dynamics and Monte Carlo methods often struggle to fully map this landscape, as they tend to become trapped in local minima and fail to capture rare conformations. This is largely due to the high energy barriers surrounding the minima or the long timescales required for conformational changes to evolve (Bernardi et al., 2015). Our goal in this work is not to design better sampling methods. Instead, we want to show how

parametrizing the low-energy subspace can facilitate problems such as sampling. We will now review some of the literature on sampling and identifying DOF.

**Space sampling.** Monte Carlo methods are advantageous for sampling configurational space due to the absence of inherent timescales, but they struggle to capture transitions between conformations and can become trapped in local minima behind high-energy barriers, leaving some regions of the energy landscape poorly sampled (Heilmann et al., 2020). Umbrella sampling, introduced by Torrie & Valleau (1977), addresses this by replacing the standard Boltzmann weighting with a biasing potential, effectively enabling a random walk across energy barriers.

Methods such as replica exchange molecular dynamics (REMD) (Hansmann & Okamoto, 1999; Sugita & Okamoto, 1999) employ MD simulations simultaneously on a series of replica systems with different conditions,e.g. temperature, and randomly exchange the states of any two replicas with a regular schedule (Qi et al., 2018) to efficiently sample and overcome high energy barriers in the landscape. However, the need for numerous replicas can significantly increase computational demands, making the approach challenging to implement in practice (Rathore et al., 2005; Liu et al., 2005; Wang et al., 2020).

Metadynamics is another approach to improve sampling of the energy landscape of a system by driving it through collective variables (CV), which represent key coordinates in the landscape (Laio & Parrinello, 2002; Bussi & Laio, 2020). In Metadynamics, Gaussian biases are periodically added to MD to prevent the system from revisiting previously explored regions, facilitating the discovery of new minima. The challenge is that finding CV is non-trivial and typically relies on prior knowledge.

**Identifying the DOF.** Identifying suitable CV for energy landscape sampling is challenging, often introducing biases and facing issues related to data quality and interpretability. Principal component analysis (PCA) has been employed to define CVs (Hori et al., 2009), but its linear nature often fails to represent the complexity of protein energy landscapes (Maisuradze et al., 2009). Recent approaches range from autoencoders learning nonlinear CVs from data (Chen & Ferguson, 2018), to path-based methods like DeepLNE (Fröhlking et al., 2024) or DeepLDA (Bonati et al., 2020; Majumder & Straub, 2024) that learn CVs from molecular dynamics trajectories. These methods have proven quite effective, but they rely on a significant amount of data and are not easily interpretable. Other including self-supervised deep neural networks, have been developed to identify slow CVs or reaction coordinates (Wehmeyer & Noé, 2018).Other CV finding methods using ML follow the coarse-graining approach to accelerate MD, improving sampling efficiency and expanding accessible phase space

(Souza et al., 2021; Majewski et al., 2023; Noé et al., 2020). Our method falls in the category of finding CVs. However, the crucial difference between our method and the above is that ours does not require trajectory or simulation data.

# 3. Theory

We will now presents our theoretical framework for discovering effective DOF, based on identifying approximate symmetries in the energy landscape. We introduce two approaches: a scalable symmetry loss function compatible with existing force-field frameworks, and a Hessian-based method effective for smaller systems. Both methods connect molecular structural dynamics to energy function symmetries, enabling systematic conformational space exploration. We derive the mathematical foundations of these approaches, showing how they lead to the discovery of physically meaningful DOF, such as dihedral angles in peptides.

## 3.1. Symmetry and DOF

While we primarily focus on physical and molecular systems, our approach is general and can be formulated in broader terms by treating the energy as a general loss function. Let $E : \mathcal{Z} \to \mathbb{R}$ represent the potential energy of a physical system, which, analogous to a loss function, we assume to be smooth over large regions of the parameter space $\mathcal{Z}$, and bounded from below. The parameters $z \in \mathcal{Z}$ correspond to the system's degrees of freedom (DOF). We assume that $\mathcal{Z}$ is a vector space. In the context of MD, the standard DOF $z = (x, p) = \{(\vec{x}_i, \vec{p}_i)\}_{i=1}^n$ includes the 3D positions $\vec{x}_i$ and momenta $\vec{p}_i = m_i d\vec{x}_i/dt$ of all particles $i \in \{1, \ldots, n\}$, where $m_i$ denotes the mass of particle $i$. Let $\vec{x}_i \in \mathcal{X}, \vec{p}_i \in \mathcal{P}$, and $\mathcal{Z} = \mathcal{X} \times \mathcal{P}$

**Temperature and kinetic energy** In this work, we are primarily interested in the static conformations of the system and therefore ignore the kinetic energy term in our formulation. By focusing on the potential energy, we capture the equilibrium properties of molecular systems. Although temperature induces thermal fluctuations in real systems, our current approach neglects these effects. These fluctuations could be incorporated into future extensions, particularly when accounting for finite-temperature effects and exploring dynamic behavior. Ignoring the kinetic DOF $\vec{p}_i$ by setting $p = 0$, our parameter space reduces to the space $\mathcal{X} \times \{0\} \subset \mathcal{Z}$ of positions $x = \{\vec{x}_i\}$. Hence, we redefine the energy to be just the **potential energy** $E : \mathcal{X} \to \mathbb{R}$.

**Lifting DOF to a group** Our core idea is that *transformations* acting on the DOF can be used to replace the original DOF, thereby lifting the DOF to a group action on $\mathcal{X}$. While not all DOF can be lifted in this manner, we demonstrate that this approach enables us to link low-energy DOF to underlying symmetries of the system. We

consider the general linear group $GL(\mathcal{X})$ acting on the parameter space $\mathcal{X}$. Global translations and rotations $SE(3)$ can also be included, but since MD potentials are generally invariant under $E(3)$, they lead to trivial symmetries, which we are not interested in here. Starting from a reference point $x_0 \in \mathcal{X}$, the orbit of $GL(\mathcal{X})$, defined as $\mathrm{Orbit}\{x = gx_0 \mid g \in GL(\mathcal{X})\}$, generates a manifold of transformed configurations. This manifold effectively describes the set of configurations related to $x_0$ by symmetry transformations. This allows us to replace $x$ with transformations $g$ that reach $x$ from $x_0$. Now, if we focus on a subset of symmetries $G \subseteq GL(\mathcal{X})$ that approximately preserves the potential energy, we can extract DOF that correspond to motion along low-energy directions, thus providing a natural way to explore the low-energy landscape of the system.

**Group parameters as DOF** In order to map $g$ to degrees of freedom, we need a parametrization for $g$. Since $GL(\mathcal{X})$ is a continuous group, we use the Lie algebra and the exponential map to parameterize $g$ in terms of the Lie algebra basis $\boldsymbol{L}_a \in \mathfrak{gl}(\mathcal{X})$, where the Lie algebra $\mathfrak{gl}(\mathcal{X}) = T_{\mathrm{id}} GL(\mathcal{X})$ is the tangent space at the identity of $GL(\mathcal{X})$. In the case of matrix Lie groups such as $GL(\mathcal{X})$, the exponential map $\exp : \mathfrak{gl}(\mathcal{X}) \to GL(\mathcal{X})$ can be written in terms of the matrix exponential. Exponentiating an element in the Lie algebra yields a group element $g = \exp(\theta \cdot \boldsymbol{L})$, where $\theta$ is a vector of parameters. In more general cases, where a group element requires a nontrivial path on the group manifold, the transformation may be expressed as a product of exponentials $g = \prod_i \exp(\theta_i \cdot \boldsymbol{L})$. In both cases, small group elements (i.e. near identity) can be expanded as $g \approx \boldsymbol{I} + \theta \cdot \boldsymbol{L} + O(\theta^2)$, where $\boldsymbol{I}$ is the identity matrix. This formulation allows us to define the group parameters $\theta$ as the new DOF, reparametrizing the system in terms of group transformations that capture the low-energy dynamics. Next, we define more concretely what we mean by low-energy dynamics and effective DOF.

## 3.2. Defining Effective Degrees of Freedom

What does it mean for this DOF to be an "effective" or "low-energy" DOF? Implicitly, low-energy DOF assumes that there exists a hierarchy in the energy scales, where some barriers in the energy landscape are much smaller than some others. In MD, we can clearly see such a hierarchy in force-fields used $E(x)$, such as AMBER (Cornell et al., 1995). The energy landscape comprises strong quadratic terms representing bond lengths and bond angles, along with nonlinear terms ("non-bonding") which are much weaker. The non-bonding energy includes Lennard-Jones ($E \sim ar^{-12} - br^{-6}$, producing the weak van der Waals forces and Hydrogen bonding) and Coulomb ($E \sim cr^{-1}$) forces, both of which dominate at short distances but decay at larger separations (App. B). The bonding energy

is convex and the energy barriers between local minima are mostly related to the non-bonding energies. Thus, effective DOF must satisfy two competing requirements:

- **At small scales**: they must provide sufficient $\delta E$ to overcome local barriers and escape metastable states, with $\delta E > E_{\text{barrier}}$.
- **At larger scales:** they must preserve the molecular structure, e.g. not break bonds, $\delta E \ll E_{\text{bond}}$.

In other words, we want the DOF to yield small $\delta E$ in a large range of parameters. When $\delta E = 0$ while moving along a DOF, we say the DOF is a *symmetry* of $E$. Let $x_0$ be a reference point at which we discover the effective DOF, and let $x = x_0 + \epsilon$. We want the DOF to be an *approximate symmetry of $E$ when $\epsilon$ is large*, but also want it to result in relatively *large $\delta E$ at small $\epsilon$* to move us straight toward energy barriers to ensure we can explore other minima. Our strategy involves two steps:

1. **Minimization:** Discover candidate set of DOF at by requiring that for large $\epsilon$:

$$S = \{\text{DOF} : |\delta E(\epsilon_l)| < \eta \quad \text{for} \quad \epsilon_l > \epsilon_{\min}\} \quad (1)$$

2. **Maximization:** Select optimal DOF:

$$\text{DOF} = \underset{\boldsymbol{L} \in S}{\arg\max} |\delta E(\epsilon_s)| \quad \text{for} \quad \epsilon_s \ll \epsilon_{\min} \quad (2)$$

where $\eta = E_{\text{barrier}}$ is a small energy scale and $\epsilon_{\min}$ corresponds to displacements on the order of non-bonded interaction distances, typically $\sim 0.1 - 0.5$nm in molecular systems. In the next section we will make these statements concrete, deriving the explicit optimization objectives.

### 3.3. Symmetry loss

Let $x \in \mathcal{X}$ be a reference point, corresponding to a configuration where the quadratic energy terms are minimized. The condition $|\delta E(\epsilon_l)| < \eta$ in the discovery step, means up to order $\eta$ changes, the $\boldsymbol{L}$ are approximate symmetries of $E$. Note that the main difference between the two steps equation 1 and equation 2 is $\epsilon_l > \epsilon_s$. Since working with exponential $g$ is difficult, we consider a small but finite transformation $g \approx I + \theta \boldsymbol{L}$, where $\theta \in \mathbb{R}$ is now just a magnitude and $\boldsymbol{L} \in \mathfrak{gl}(\mathcal{X})$, yielding

$$\delta E \approx \theta \nabla E(x) \cdot \boldsymbol{L}x \quad (3)$$

The optimization objective corresponding to step 1 of our method is to find the $\boldsymbol{L}$ that minimizes the symmetry loss

$$\textbf{Symmetry loss:} \quad \mathcal{L}(\boldsymbol{L}, x) = (\nabla E(x) \cdot \boldsymbol{L}x)^2 \quad (4)$$

subject to $\|\boldsymbol{L}\|_F \leq 1$, ensuring that $\boldsymbol{L}$ remains within a bounded region of the Lie algebra. The $\boldsymbol{L}$ minimizing equation 4 define the set of **effective DOF**. Note that $\boldsymbol{L}$ are not necessarily exact symmetries of $E$ and the symmetry loss does not need to vanish (e.g. in MD $\delta E \sim O(E_{\text{nonbond}})$ is permissible). However, in problems such as MD there are important global symmetry considerations, which we discuss next.

### 3.4. Excluding global symmetries

In MD, the configuration space $\mathcal{X}$ is naturally isomorphic to $\mathbb{R}^{n \times d}$, where $n$ is the number of particles and $d$ is the spatial dimension. For $d = 3$, the system often has a global $SE(3)$ symmetry, corresponding to rotations and translations in three-dimensional space. However, we are not interested in this symmetry, as it represents trivial motions that do not affect the relative configuration of the particles. Therefore, we restrict the Lie algebra element $\boldsymbol{L}$ to act on the particle indices, while being invariant under $SE(3)$ transformations.

Given this restriction, the action of $\boldsymbol{L}$ affects only the $n$-dimensional part of $x \in \mathbb{R}^{n \times d}$. The condition for approximate symmetry, $(\nabla E \cdot \boldsymbol{L}x)^2$, can now be written as

$$SE(3)\textbf{-invariant loss:} \quad\quad\quad (5)$$

$$\mathcal{L}(\boldsymbol{L}, x) = \left(\sum_{i,j,\mu} \frac{\partial E}{\partial x_j^\mu} \boldsymbol{L}_j^i x_i^\mu\right)^2 = \left(\text{Tr}\left[(\nabla E)^\top \boldsymbol{L}x\right]\right)^2$$

where $i, j$ index the particles, and $\mu$ indexes the spatial components. Here, $\nabla E \in \mathbb{R}^{n \times d}$ is the gradient of the energy with respect to the particle positions, and the matrix product involves $\boldsymbol{L}$, which acts on the particle indices, while $x$ is the current configuration of the system. We will be working with equation 5 instead of equation 4. Additionally, in small molecular systems we can use another level of simplification using the Hessian, described next.

### 3.5. Hessian Approach for Symmetry Loss

Assume that $x_*$ is a critical point of the energy, meaning that $\nabla E(x_*) = 0$. The $x$ in equation 5 is not necessarily a critical point, but we can assume $x$ is *close* to a $x_*$, meaning $x = x_* + \epsilon$, where $\epsilon \sim \mathcal{N}(0, \sigma^2 I)$ is a random perturbation around $x_*$. We can Taylor expand $\nabla E(x)$ around $x_*$ to first order

$$\nabla E(x) \approx \nabla E(x_*) + H(x_*) \cdot (x - x_*) = H(x_*) \cdot \epsilon \quad (6)$$

Here, $H(x_*)$ is the Hessian matrix at $x_*$, which has components $H_{\mu\nu}^{ij} = \partial^2 E / \partial x_i^\mu \partial x_j^\nu$.

**Expectation over Gaussian Perturbations** Now, assume that we have many samples $x$, such that $\epsilon$ is drawn from a

Gaussian distribution. Since $\epsilon \sim \mathcal{N}(0, \sigma^2 I)$, we have

$$\mathbb{E}[\epsilon_i^\mu \epsilon_k^\nu] = \sigma^2 \delta_{ik} \delta^{\mu\nu} \tag{7}$$

Substituting this expansion into the symmetry loss

$$\mathbb{E}[\mathcal{L}(\boldsymbol{L}, x)] = \mathbb{E}\left[\left(\mathrm{Tr}\left[\epsilon^\top H(x_*)^\top \boldsymbol{L}(x_* + \epsilon)\right]\right)^2\right] \tag{8}$$

where $H = H(x_*)$ and we used the symmetry of the Hessian ($H^\top = H$). Note that in the *Minimization* step equation 1 we want the approximate symmetry condition to hold for $\epsilon$ which are relatively large compared to the *Maximization* step equation 2. To distinguish between the $\delta E$ of these two steps, we should choose different orders of Taylor expansion for them. Therefore, for the minimization step we will keep the $O(\epsilon^2)$ terms, but for the maximization we only keep $O(\epsilon)$.

The cross-term in equation 8 is $O(\epsilon^3)$ and vanishes because $\epsilon$ is normal. Since $H$ and $\boldsymbol{L}$ always appear together and for ease of notation, let us denote $\boldsymbol{K} \equiv H\boldsymbol{L}$. For the first term in equation 8, using equation 7 we get (see Appendix A.1 for all derivations below)

$$\mathbb{E}\left[\mathrm{Tr}\left[\epsilon^\top \boldsymbol{K} x_*\right]^2\right] = \sigma^2 \|\boldsymbol{K} x_*\|^2 \tag{9}$$

The last term, this in equation 8 yields

$$\mathbb{E}\left[\mathrm{Tr}\left[\epsilon^\top \boldsymbol{K}\epsilon\right]^2\right] = \sigma^4\left(2\mathrm{Tr}\left[\boldsymbol{K}_S^2\right] + \mathrm{Tr}\left[\boldsymbol{K}_S\right]^2\right) \tag{10}$$

where $\boldsymbol{K}_S = (\boldsymbol{K}^\top + \boldsymbol{K})/2$ is the symmetric part of $\boldsymbol{K}$. Putting these together, the symmetry loss in this approximation becomes

**Hessian symmetry loss:** (11)

$$\mathbb{E}\left[\mathcal{L}(\boldsymbol{L}, x)\right] \approx \sigma^2 \|\boldsymbol{K} x_*\|^2 + \sigma^4\left(2\mathrm{Tr}\left[\boldsymbol{K}_S^2\right] + \mathrm{Tr}\left[\boldsymbol{K}_S\right]^2\right)$$

### 3.6. Analytical Solutions to the Trace Loss in 1D

We note that there exists a simple analytical way to minimize each of the two loss terms. In the 1-D case, the $\sigma^4$ term only depends on the eigenvalues of the symmetric part of $\boldsymbol{K}_S = H\boldsymbol{L} + \boldsymbol{L}^\top H$. Let $\lambda_1, \lambda_2, \ldots, \lambda_n$ be the eigenvalues, then the optimization is equivalent to minimizing $\sum_{i=1}^n 2\lambda_i^2 + \left(\sum_{i=1}^n \lambda_i\right)^2$ which can be minimized by minimizing the operator norm of $\boldsymbol{K}_S$. The space of $\boldsymbol{L}$ minimizing the operator norm of $K_S$ can be given as a combination of the following (see Appendix A for details) :

### 1. Symmetric $L$ part:

**Proposition 3.1** ($\boldsymbol{L}$ in "slow" subspace of the Hessian). *If $\boldsymbol{L}$ has support only on the span of eigenvectors with the smallest eigenvalues of $H^2$*

$$\|H\boldsymbol{L}\|^2 = \mathrm{Tr}\left[\boldsymbol{L}^T H^2 \boldsymbol{L}\right] \tag{12}$$

*then it minimizes the $\sigma^4$ term in equation 11 and also minimizes* $\mathrm{Tr}\left[\boldsymbol{K} x\right]^2$ *for arbitrary $x$.*

In practice, we are content with having small but nonzero $\mathrm{Tr}\left[\boldsymbol{L}^T H^2 \boldsymbol{L}\right]$. In this case, $\boldsymbol{L}$ corresponds to directions in the configuration space along which the energy is approximately flat.

### 2. Anti-Symmetric $L$ part:

**Proposition 3.2** (Anti-symmetric $\boldsymbol{K}$). *If $H\boldsymbol{L}$ is antisymmetric, the trace loss can also be minimized. This requires*

$$H\boldsymbol{L} + \boldsymbol{L}^\top H = 0 \quad \Rightarrow \quad \mathrm{Tr}\left[\boldsymbol{K}_S\right] = 0, \quad \mathrm{Tr}\left[\boldsymbol{K}_S^2\right] = 0$$

*which implies that $\boldsymbol{L}$ generates transformations that preserve the structure of the Hessian. One solution to this condition is if $\boldsymbol{L}$ itself is antisymmetric $\boldsymbol{L}^\top = -\boldsymbol{L}$. In this case, the commutator between $\boldsymbol{H}$ and $\boldsymbol{L}$ vanishes*

$$[\boldsymbol{L}, H] = 0$$

*This implies that $\boldsymbol{L}$ commutes with $H$, and therefore defines symmetry directions where the Hessian is invariant.*

If $H$ has a degenerate subspace corresponding to $k$ degenerate eigenvalues, this subspace has an inherent $SO(k)$ symmetry. Because of this, the Lie algebra elements $\boldsymbol{L} \in \mathfrak{so}(k)$ of this subspace symmetry satisfy $[\boldsymbol{L}, H] = 0$. This is a special case of the proposition 3.2. More formally:

**Proposition 3.3** (Degenerate subspace solution). *let $\Lambda$ be the diagonalized form of $H$, with $Q\Lambda Q^\top = H$. If $\Lambda$ has a set of $k$-fold degenerate eigenvalues $\lambda_1 = \lambda_2 = \cdots = \lambda_k$, the corresponding eigenspace forms a $k$-dimensional subspace of symmetry. The action of $\boldsymbol{L}$ in this subspace can be viewed as a rotation, and $\boldsymbol{L}$ can be chosen to belong to the Lie algebra of rotations $SO(k)$ restricted to the degenerate subspace:*

$$\boldsymbol{L} \in \mathfrak{so}(k), \quad \boldsymbol{L}^\top = -\boldsymbol{L} \tag{13}$$

*The matrix $\boldsymbol{L}$ generates rotations within the degenerate eigenspace, leaving the overall structure of $H$ invariant.*

In the three dimensional case, owing to the global spatial symmetries, we focus on $\boldsymbol{L} \in \mathbb{R}^{n \times n}$ which only act on the particle indices $(i, j)$ and not the spatial indices $(\mu, \nu)$. Thus, we can relate this anti-symmetric one dimensional solution to the general case by having $[H, \boldsymbol{L}] = 0$ be given as $[H_{\mu\nu}, \boldsymbol{L}] = 0$ for all $(\mu, \nu)$. This is not always possible to do and therefore we approximate it to the condition $[\boldsymbol{H}_2, \boldsymbol{L}] = 0$ for $\boldsymbol{H}_2 = \sum_\mu H_{\mu,\mu}$. Similarly for the symmetric case, we replace $\|H\boldsymbol{L}\|^2$ with $\|\boldsymbol{H}_2 L\|^2$ for $\boldsymbol{H}_2 = \sum_{\mu,\nu} H_{\mu,\nu}^2$. Thus, we generalize this one-dimensional analytic framework by looking at slow eigenspaces and degenerate eigenspaces for suitable $SE(3)$ invariant matrices $\boldsymbol{H}_2$.

## 4. Selecting optimal effective DOF

Using the discovery procedure, we get a set of Lie algebra elements $L_a$. Given the set of $L_a$, we now want to find the transformations that help us navigate the energy landscape most effectively. Unlike the discovery step for $L$, where we wanted $\delta E$ to be small, now we want to maximize it as in equation 2. Therefore, we define the most effective $L$ as the one leading to the largest perturbations in energy. As stated above, we want the largest $\delta E$ in the close vicinity of $x_*$, meaning this time we will assume $\epsilon$ is very small and will only keep the $\sigma^2$ term, discarding the $\sigma^4$. Using the result from equation 9 as a proxy of the magnitude of structural change, we can solve the following optimization problem to get the most effective DOFs among $n_L$ discovered $L$:

$$L(c) \equiv \sum_{a=1}^{n_L} c_a L_a,$$

$$\text{Find } c_* = \underset{c, \|c\|_2 = 1}{\arg\max} \|H L(c) x_*\|^2 \qquad (14)$$

But since this term is identical to the $\sigma^2$ term we used for the discovery of $L$, it makes sense to only minimize the $\sigma^4$ term for discovering candidate $L$ and use the $\sigma^2$ term only in the second step. This is what we do in our experiments. In the end, we keep the top two $L(c)$ as the effective DOF. When using equation 4 for discovering $L$, we also use the full $H_{\mu\nu}^{ij}$ in equation 14. When using the other Hessian approaches, we use $H_2$ instead of $H$ in equation 14.

**Optimal $L$ without Hessian.** For the optimization version of the problem, we see that as mentioned before the $L$ are discovered at high noise levels (large $\sigma$) by minimizing the symmetry loss over the samples. In order to find the most effective DOF, we maximize the symmetry loss at low noise levels (small $\sigma$). Given $m$ samples $\epsilon_j \sim \mathcal{N}(0, \sigma_{\text{eff}} \mathbf{I}^{n \times d})$ where $\sigma_{\text{eff}}$ is smaller than $\sigma$ used for discovery. Then

$$c_* = \underset{c \in \mathbb{R}^K, \|c\|_2 = 1}{\arg\max} \sum_{j=1}^{m} \text{Tr} \left[ \nabla E(x_* + \epsilon_j)^\top L(c)(x_* + \epsilon_j) \right]^2 \qquad (15)$$

## 5. Experiments

To evaluate our method's effectiveness, we conducted experiments on two well-characterized molecular systems: alanine dipeptide and the designed mini-protein chignolin. These systems serve as canonical test cases in the molecular dynamics community, offering a balance between computational tractability and biological relevance.

In the case of the alanine dipeptide, the two DOF that capture the most important conformations are known to be the dihedral angles $\phi$ and $\psi$ over the peptide bonds.

$(\phi, \psi)$ are used as coordinates to describe the states of the molecule as a density plot called the Ramachandran plot. We want to investigate whether the DOF discovered using our method can be used to effectively explore the states of the system. We will examine different regions of the Ramachandran plot reached by varying our discovered DOF and whether we can discover the known conformers of the alanine dipeptide in this way. We will also investigate whether $(\phi, \psi)$ overlap with our DOF, which are discovered directly from the forcefield without using any prior knowledge about the importance of the peptide bonds. In case of chignolin, there are a lot of local minima around a properly folded state and another near a misfolded state (forming two large groups of conformers). In the literature (Kührová et al., 2012), it has been shown that the two groups of conformers can be distinguished by the configuration of the carbon backbone of the Glycine 7 residue in chignolin. Denoting the backbone dihedral angles of Gly7 as $\phi, \psi$, we repeat the same analysis for chignolin. However, we only examine chignolin in the presence of solvent as the $\beta$ hairpin structure of chignolin is only formed in presence of solvent.

As the conformers of the alanine dipeptide as well as the ramachandran plot change with the ambient medium, we consider two MD setting for alanine dipeptide. For chignolin the $\beta$ hairpin configuration is only observed in solvent, so we only consider chignolin in water. In total, we consider three MD settings. 1) **Alanine Dipeptide in Vacuum** where we only use the amber forcefield (amber99sbnmr) corresponding to interactions within the molecule and 2) **Alanine Dipeptide in Water** where we use the molecule forcefield (amber99sbnmr) along with the amber forcefield for the solvent (amber99_obc) modelled as implicit. 3) **Chignolin in Water** where we use the molecule forcefield (amber99sbnmr) along with the amber forcefield for the solvent (amber99_obc) modelled as implicit. Additionally, we put Hbond constraints (that keeps the length of bonds between heavy atoms and hydrogen fixed) and use heavy hydrogen in order to stabilize the integration steps (as used in (Satoh et al., 2006)).

**Methods for Extracting DOF:** Based on our theoretical results, we will use four methods to discover effective DOF:

1. **Direct Optimization:** Solving equation 5

2. **Full Hessian:** Minimizing the $\sigma^4$ in equation 11

3. **Slow Hessian:** Using smallest eigenvalue subspace of $H_2$ (minimizng the $\sigma^2$ term of equation 11 for arbitrary $x_*$).

4. **Degenerate Hessian:** Using a degenerate subspace of $H_2$. We use the subspace with the highest dimensions

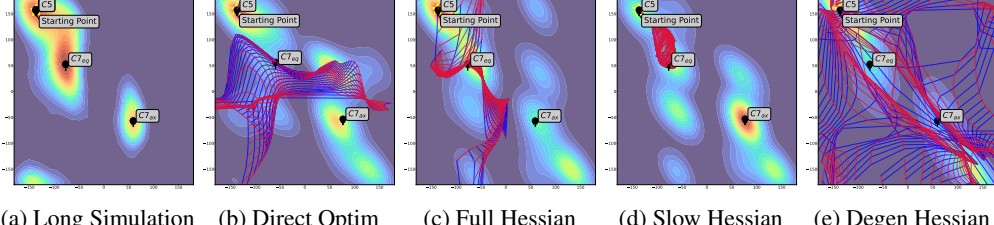

(a) Long Simulation    (b) Direct Optim    (c) Full Hessian    (d) Slow Hessian    (e) Degen Hessian

Figure 1: **Ramachandran plot for alanine dipeptide in vacuum:** a) long (500ns) simulation starting at $\beta$, b) direct optimization of equation 4, c) analytically solving the $\sigma^4$ term in equation 11 d) slow subspace of $H_2$ e) fast degenerate subspace of $H_2$. The blue and red grid lines are traced by transforming $\beta$ alanine along the discovered DOF.

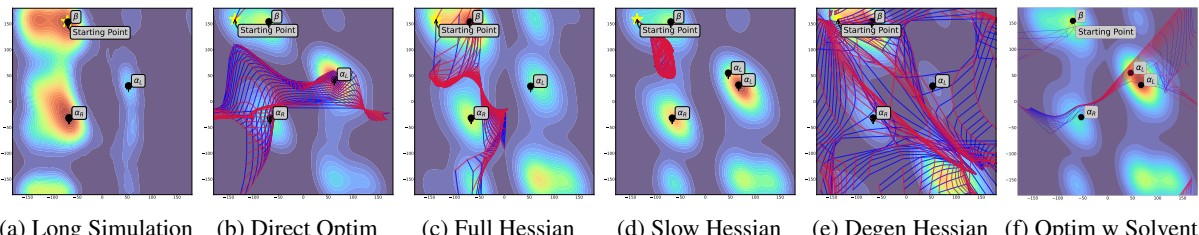

(a) Long Simulation    (b) Direct Optim    (c) Full Hessian    (d) Slow Hessian    (e) Degen Hessian    (f) Optim w Solvent

Figure 2: **Ramachandran plot for alanine dipeptide in water:** a) long (500ns) openMM sim with implicit solvent starting at $\beta$ alanine dipeptide, b) direct optimization of equation 4 c) $\sigma^4$ analytical solution d) slow subspace of $H_2$ e) fast degenerate subspace of $H_2$ f) direct optimization over the forcefield with solvent. The blue and red grid lines are transforming $\beta$ alanine along the DOF.

Note that the methods involving an explicit Hessian cannot be easily applied to settings with a stochastic solvent force. But, the direct optimization of equation 4 is applicable in all cases, including used systems with an implicit model for the solvent. We discuss the results for both settings: optimization method using just intra-molecular forces and optimization method including solvent forcefield.

**Procedure:** Our experimental procedure is as follows:

1. Start from an initial conformation $x_0$, the $C_5$ conformer for alanine dipepetide and a properly folded conformer in case of chignolin.
2. Extract the top two DOF $L_1, L_2$ near $x_0$.
3. Make a $31 \times 31$ grid of angles $(\theta_1, \theta_2) \in [0, 2\pi)^2$.
4. Generate deformed conformers $x = e^{\theta_1 L_1 + \theta_2 L_2} x_0$.
5. Use `openmm.minimize` on $x$ and run short $2ps$ simulations to find stable conformations from the deformed structure.

We then compare the conformers found using the method above against conformers found using a baseline method of sampling. For our baseline we run long openMM simulations for $500ns$ at $300K$ with friction coefficient of $1ps^{-1}$ and step size of $2fs$ amounting to $2.5e8$ steps. We use the same amber forcefields in both the last step of our method and the baseline simulations in order to maintain consistency.

**Comment on sampled densities.** It is important to note that our goal is *not* recovering the density of states found via the long simulation. In fact, our goal is to be able to visit states which are much harder to reach using the baseline method. Figures 1–3 all show the density of states sampled. We have also plotted the energies of the states (Appendix C), confirming that we indeed arrive at low-energy states.

### 5.1. Experiments in Vacuum

When modeling alanine dipeptide in vacuum, we only consider the molecular forces between the atoms. For this setting, we see that all the Hessian-based methods recover all the major conformations of alanine dipeptide with relatively short simulation times. As the Lie algebra elements $L$ in our problem span a $\mathbb{R}^{n^2}$ space, we need at least $O(n^2)$ point to avoid overfitting. For our experiments, we use $16n^2$ samples for discovery and $16n^2$ samples for finding the most effective degrees of freedom. Using larger values of $\epsilon$ can give us more information about long-range symmetries but using large $\epsilon$ also increases the stochasticity causing very high variance in the estimates.

### 5.2. Experiments with Solvent

We experimented with both alanine dipeptide (Figs 2, 4b) and Chignolin (Figs 3, 5) in water. Alanine dipeptide has different stable conformers in water than in vacuum, suggesting that incorporating the solvent force is essential for

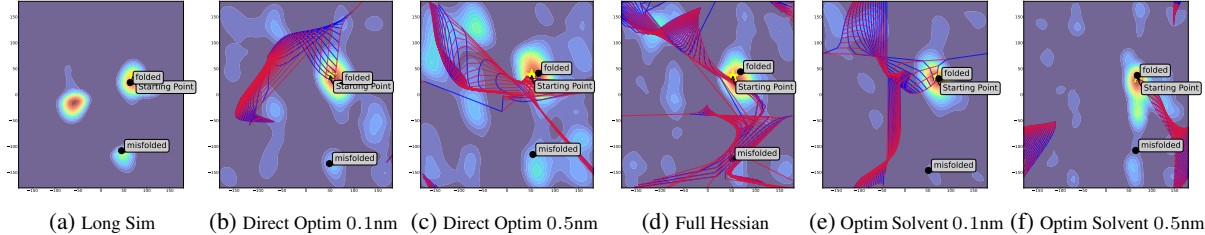

| (a) Long Sim | (b) Direct Optim 0.1nm | (c) Direct Optim 0.5nm | (d) Full Hessian | (e) Optim Solvent 0.1nm | (f) Optim Solvent 0.5nm |

Figure 3: **Ramachandran plot for chignolin in water** a) long sim (500ns) with implicit solvent starting at folded chignolin, b), c) direct optimization with DOF discovered at medium (0.1nm) and long range (0.5nm), d) analytical solution of $\sigma^4$ term in equation 11 e),f) direct optimization of forcefield with solvent and DOF discovered at medium (0.1nm) and long range (0.5nm). The blue and red grid lines are transforming chignolin along the discovered DOF.

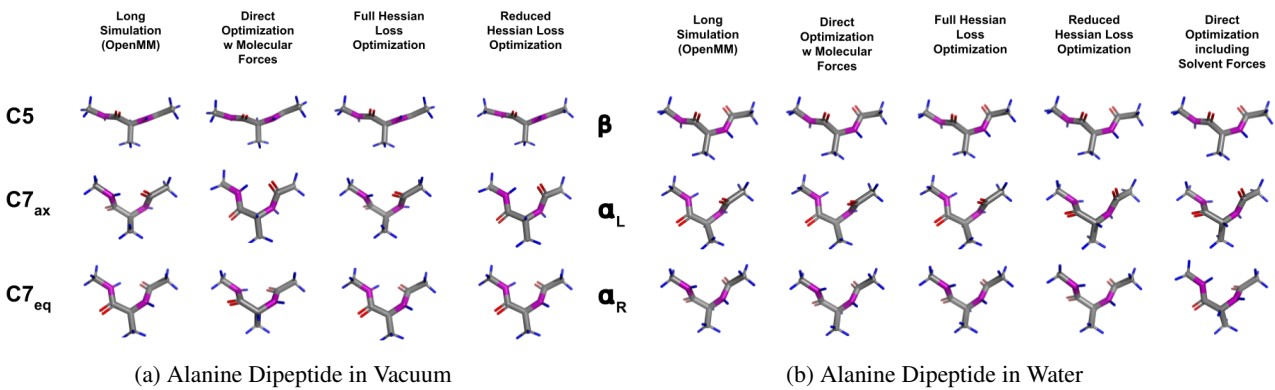

| (a) Alanine Dipeptide in Vacuum | (b) Alanine Dipeptide in Water |

Figure 4: Closest Structure to the known conformers. We only consider a structure to be a conformer candidate if it is stable and its $(\phi, \psi)$ dihedral angles are close to that of the corresponding conformer.

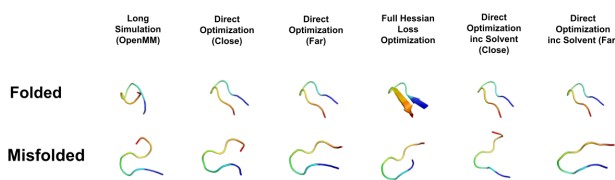

Figure 5: Closest Structure to the conformers of Chignolin discovered for simulations in water. Only the last two columns use the solvent force field to calculate the DOF.

modeling the DOF. Note that our optimization approach using equation 4 can use the forcefield with explicit solvent forces included to calculate the DOF, and can therefore be applied to the case with solvent. However, our Hessian-based methods do not work with stochastic forces, and therefore we use the DOF discovered using the forcefield in vacuum also in the case with solvent. Our results suggest the vacuum DOF also work well in the presence of solvent. Similarly, we also use the Hessian based method as well as the direct optimization method to calculate DOF for chignolin in vacuum and use those to navigate the free energy landscape in the presence of solvent. Moreover, we also see that although the Hessian-based approaches do not easily transfer over to the solvent scenario, the optimization

based approach can be easily generalized to any forcefield. So, we also incorporate the implicit solvent forcefield in the optimization-based approach to find symmetries over the combined forcefields. As seen in the figures above, all the approaches proposed discover the major conformers of alanine dipeptide as well as folded and misfolded conformers of chignolin. Another important fact that we want to highlight is that discovered DOF are general enough to allow us to find the conformers in settings with forcefields slightly different from the ones they were learnt on.

**Discussion on speed**

Although our methods use the openmm simulation in the final step, we only need a fraction of the total steps required by openmm to find all conformations. Furthermore, the simulations can be run for all grid points simultaneously. Thus, in principle, the effective time required for simulations using our method could be orders of magnitude less than the time required for the baseline long openmm simulations. The major bottleneck for our approach is the Hessian computation which scales quadratically with the size of the system (2sec for alanine-dipeptide, 200sec for chignolin) and does not easily generalize to stochastic forcefields. While the direct optimization method overcomes

|  |  | **0.25** | **0.5** | **2.5** | **5** | **25** | **50** | **250** | **500** |
|---|---|---|---|---|---|---|---|---|---|
| Alanine Dipeptide in Vacuum | $C5$ | 0.257 | 0.268 | 0.233 | 0.247 | 0.246 | 0.238 | 0.215 | 0.23 |
|  | $C7_{eq}$ | 0.479 | 0.429 | 0.448 | 0.436 | 0.436 | 0.425 | 0.382 | 0.408 |
|  | $C7_{ax}$ | 0 | 0 | 0 | 0 | 0 | 0.027 | 0.117 | 0.059 |
| Alanine Dipeptide in Solvent | $\beta$ | 0.411 | 0.424 | 0.416 | 0.404 | 0.367 | 0.39 | 0.39 | 0.39 |
|  | $\alpha_L$ | 0 | 0 | 0 | 0 | 0.001 | 0.001 | 0.001 | 0.001 |
|  | $\alpha_R$ | 0.043 | 0.043 | 0.048 | 0.047 | 0.041 | 0.043 | 0.043 | 0.043 |
| Chignolin in Solvent | folded | 0.992 | 0.994 | 0.997 | 0.79 | 0.206 | 0.4 | 0.4 | 0.4 |
|  | misfolded | 0 | 0 | 0 | 0.183 | 0.073 | 0.048 | 0.048 | 0.048 |

Table 1: Fractional Distribution of Conformers with respect to elapsed simulation time (in ns) for a standard long openmm simulation. This denotes the relative frequency and exploration of the conformal landscape as a function of the time elapsed in the simulation.

|  |  | **0.002** | **0.01** | **0.02** | **0.1** | **0.2** | **1** | **2** | **10** |
|---|---|---|---|---|---|---|---|---|---|
| Alanine Dipeptide in Vacuum | $C5$ | 0.4659 | 0.3676 | 0.2463 | 0.2168 | 0.2264 | 0.2082 | 0.2232 | 0.2340 |
|  | $C7_{eq}$ | 0.1818 | 0.2745 | 0.3515 | 0.3745 | 0.3708 | 0.3311 | 0.3357 | 0.3367 |
|  | $C7_{ax}$ | 0 | 0 | 0 | 0 | 0 | 0.0768 | 0.0467 | 0.0226 |
| Alanine Dipeptide in Solvent | $\beta$ | 0.9205 | 0.7451 | 0.6052 | 0.4808 | 0.4336 | 0.4359 | 0.4422 | 0.4505 |
|  | $\alpha_L$ | 0 | 0 | 0 | 0 | 0 | 0.0017 | 0.0063 | 0.0061 |
|  | $\alpha_R$ | 0.0000 | 0.0956 | 0.2166 | 0.3079 | 0.3417 | 0.3383 | 0.3238 | 0.3200 |

Table 2: Fractional Distribution of Conformers with respect to elapsed simulation time (in ns) for a REMD simulation with 8 parallel simulations at logarithmically spaced out temperatures between 300 K - 500 K with an attempted transition between adjacent temperatures every 50 steps. This denotes the relative frequency and exploration of the conformal landscape as a function of the time elapsed in the simulation.

the second challenge in principle we find it to be less stable than the Hessian-based approaches in practice and it also has similar quadratic computation cost (15sec for alanine-dipeptide and 450sec for chignolin). The pairwise nature of the atomic interactions leads to quadratic cost, but it can be pruned to be almost linear by using distance cut-off. Table 1 compares the relative distribution of different conformers as a function of the simulation time. This shows that the conformational landscape is poorly explored by a long simulation. On the other hand, our proposed method is able to find a structure close to every conformer within 2 ps of simulation time leading to much more efficient and thorough exploration of the conformational landscape. Additionally we also compare it to Replica Exchange Meta Dynamics on Alanine-dipeptide using 8 parallel simulations at logarithmically spaced out temperatures between 300 K - 500 K with an attempted transition between adjacent temperatures every 50 steps. Table 2 shows that even under this enhanced sampling formulation, the proposed method produces orders of magnitude in speed-up (as some conformers are only discovered after 1 ns of simulation time).

## Acknowledgements

We thank the anonymous ICML reviewers for helpful feedback and discussions. Jeet Mohapatra was partly from NSF Expeditions grant (award 1918839): Collaborative Research: Understanding the World Through Code. Csaba Both's work was done partly during his internship at the MIT-IBM Watson AI Lab.

## Impact Statement

This paper presents work whose goal is to advance the field of Machine Learning. There are many potential societal consequences of our work, none which we feel must be specifically highlighted here.

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

# A. Derivations Of Results

## A.1. Hessian Symmetry Loss

Substituting this expansion into the symmetry loss

$$
\begin{aligned}
\mathbb{E}[\mathcal{L}(\boldsymbol{L}, x)] &= \mathbb{E}\left[\left(\text{Tr}\left[\epsilon^\top H(x_*)^\top \boldsymbol{L}(x_* + \epsilon)\right]\right)^2\right] \\
&= \mathbb{E}\left[\text{Tr}\left[\epsilon^\top H \boldsymbol{L} x_*\right]^2 + 2\text{Tr}\left[\epsilon^\top H \boldsymbol{L} x_*\right]\text{Tr}\left[\epsilon^\top H \boldsymbol{L}\epsilon\right] + \text{Tr}\left[\epsilon^\top H \boldsymbol{L}\epsilon\right]^2\right]
\end{aligned}
\tag{16}
$$

where $H = H(x_*)$ and we used the symmetry of the Hessian ($H^\top = H$). For the second term we have

$$
\mathbb{E}\left[\text{Tr}\left[\epsilon^\top H \boldsymbol{L} x_*\right]\text{Tr}\left[\epsilon^\top H \boldsymbol{L}\epsilon\right]\right] = 0 \quad \text{(vanishes due to } \mathbb{E}[\epsilon^3] = 0) \tag{17}
$$

Since $H$ and $\boldsymbol{L}$ always appear together and for ease of notation, let us denote $\boldsymbol{K} \equiv H\boldsymbol{L}$. For the first term in equation 16, using equation 7 we get

$$
\mathbb{E}\left[\text{Tr}\left[\epsilon^\top \boldsymbol{K} x_*\right]^2\right] = \sum_{i,j,\mu,\nu} \mathbb{E}\left[\epsilon_i^\mu \epsilon_j^\nu\right](\boldsymbol{K}x_*)_\mu^i (\boldsymbol{K}x_*)_\nu^j \tag{18}
$$

$$
= \sigma^2 \sum_{i,\mu} (\boldsymbol{K}x_*)_\mu^i (\boldsymbol{K}x_*)_\mu^i = \sigma^2 \|\boldsymbol{K}x_*\|^2 \tag{19}
$$

The last term, this in equation 16 yields

$$
\mathbb{E}\left[\text{Tr}\left[\epsilon^\top \boldsymbol{K}\epsilon\right]^2\right] = \sum_{i,j,k,l,\mu,\nu,\rho,\lambda} \boldsymbol{K}_{\mu\nu}^{ij} \boldsymbol{K}_{\rho\lambda}^{kl} \mathbb{E}\left[\epsilon_i^\mu \epsilon_j^\nu \epsilon_k^\rho \epsilon_l^\lambda\right] \tag{20}
$$

$$
= \sigma^4 \sum_{i,j,\mu,\nu} \left\{\boldsymbol{K}_{\mu\nu}^{ij}\left[\boldsymbol{K}_{\mu\nu}^{ij} + \boldsymbol{K}_{\nu\mu}^{ji}\right] + \boldsymbol{K}_{\mu\mu}^{ii} \boldsymbol{K}_{\nu\nu}^{jj}\right\} \tag{21}
$$

$$
= \sigma^4 \left\{\text{Tr}\left[\boldsymbol{K}\left(\boldsymbol{K}^\top + \boldsymbol{K}\right)\right] + \text{Tr}\left[\boldsymbol{K}\right]^2\right\}
$$

$$
= \sigma^4 \left\{\frac{1}{2}\text{Tr}\left[\left(\boldsymbol{K}^\top + \boldsymbol{K}\right)^2\right] + \text{Tr}\left[\boldsymbol{K}\right]^2\right\} \tag{22}
$$

Defining the symmetric part $\boldsymbol{K}_S = (\boldsymbol{K}^\top + \boldsymbol{K})/2$, we have

$$
\mathbb{E}\left[\text{Tr}\left[\epsilon^\top \boldsymbol{K}\epsilon\right]^2\right] = \sigma^4 \left\{2\text{Tr}\left[\boldsymbol{K}_S^2\right] + \text{Tr}\left[\boldsymbol{K}_S\right]^2\right\} \tag{23}
$$

where we used the fact that $\text{Tr}\left[\boldsymbol{K}\right] = \text{Tr}\left[\boldsymbol{K}_S\right]$. Putting these together, the symmetry loss in this approximation becomes

**Hessian symmetry loss:**
$$
\mathbb{E}\left[\mathcal{L}(L, x)\right] \approx \sigma^2 \|\boldsymbol{K}x_*\|^2 + \sigma^4 \left\{2\text{Tr}\left[\boldsymbol{K}_S^2\right] + \text{Tr}\left[\boldsymbol{K}_S\right]^2\right\}. \tag{24}
$$

Now, note that since $\boldsymbol{K} \equiv H\boldsymbol{L}$ the components are $\boldsymbol{K}_{\mu\nu}^{ij} = \sum_k H_{\mu\nu}^{ik} \boldsymbol{L}^{kj}$. Let us define the trace over spatial indices, $\mu, \nu$, and the node (particle) indices $i, j$ as follows

$$
\text{Tr}_\text{s}\left[H\right]^{ij} \equiv \sum_\mu H_{\mu\mu}^{ij}, \qquad\qquad \text{Tr}_\text{n}\left[H\right]_{\mu\nu} \equiv \sum_i H_{\mu\nu}^{ii} \tag{25}
$$

For the trace terms we have

$$
\text{Tr}\left[\boldsymbol{K}\right] = \sum_{i,\mu} \boldsymbol{K}_{\mu\mu}^{ii} = \sum_{i\mu} H_{\mu\mu}^{ik} \boldsymbol{L}^{ki} = \text{Tr}_\text{n}\left[\text{Tr}_\text{s}\left[H\right]\boldsymbol{L}\right]
$$

$$
\text{Tr}\left[\boldsymbol{K}\boldsymbol{K}^\top\right] = \sum_{ij\mu\mu}\left(\boldsymbol{K}_{\mu\nu}^{ij}\right)^2 = \sum_{i\mu} H_{\mu\nu}^{ik} \boldsymbol{L}^{kj} H_{\mu\nu}^{il} \boldsymbol{L}^{lj} = \text{Tr}_\text{n}\left[\boldsymbol{L}^\top \text{Tr}_\text{s}\left[H^2\right]\boldsymbol{L}\right]
$$

$$
\text{Tr}\left[\boldsymbol{K}^2\right] = \sum_{ij\mu\mu}\left(\boldsymbol{K}_{\mu\nu}^{ij}\right)^2 = \sum_{i\mu} H_{\mu\nu}^{ik} \boldsymbol{L}^{kj} H_{\nu\mu}^{jl} \boldsymbol{L}^{li} = \text{Tr}_\text{s}\left[\text{Tr}_\text{n}\left[HLHL\right]\right] \tag{26}
$$

This provides a relationship between the Hessian, the Lie algebra element $\boldsymbol{L}$, and the effective DoF defined by $\boldsymbol{L}$, allowing us to identify approximate symmetries by minimizing this loss. Note that the above still approximately holds even $x_*$ is not a critical point, but some point where the gradient is small, meaning $|\nabla E(x_*)| < \eta$ for some small $\eta$.

## A.2. Analytical Solutions to the Trace Loss in 1D

In the 1-D case, the $\sigma^4$ term only depends on the eigenvalues of the symmetric part of $\boldsymbol{K}_S = \boldsymbol{HL} + \boldsymbol{L}^\top \boldsymbol{H}$. Let $\lambda_1, \lambda_2, \ldots, \lambda_n$ be the eigenvalues, then the optimization is equivalent to minimizing $\sum_{i=1}^n 2\lambda_i^2 + \left(\sum_{i=1}^n \lambda_i\right)^2$ which can be minimized by minimizing the operator norm of $\boldsymbol{K}_S$. This can be equivalently written as

$$x^T \left(\boldsymbol{L}^T \boldsymbol{H} + \boldsymbol{HL}\right) x = x^T \left(\boldsymbol{H}^T + \boldsymbol{H}\right) \boldsymbol{L}x = O(\varepsilon) \tag{27}$$

Any $\boldsymbol{L}$ that satisfies equation 27 for arbitrary $x$ is a symmetry of the Hessian. We establish the following result, that shows that the slow modes of the Hessian form and the rotation of the degenerate subspaces of the Hessian are the most dominant modes of the symmetry.

**Theorem A.1.** *For a given symmetric real matrix $\boldsymbol{A}$ with eigen-decomposition given as $\boldsymbol{A} = \boldsymbol{V}\Lambda\boldsymbol{V}^T$, we see that only matrices $\boldsymbol{B}$ of the form*

$$\boldsymbol{B} = \boldsymbol{V}\Gamma\boldsymbol{V}^T, \ \text{with} \ \sum_{ij} (\lambda_j \Gamma_{ji} + \lambda_i \Gamma_{ij})^2 \leq O_r(\epsilon^2)$$

*satisfy the equation*

$$\forall x \ \boldsymbol{s.t} \ \|x\|_2 = 1, \quad x^T \left(\boldsymbol{B}^T \boldsymbol{A} + \boldsymbol{AB}\right) x \leq O(\epsilon) \tag{28}$$

*Proof.* For any real symmetric matrix $\boldsymbol{A}$, there exists an eigendecomposition for $\boldsymbol{A}$ such that $\boldsymbol{A} = \boldsymbol{V}\Lambda\boldsymbol{V}^T$ where $\boldsymbol{VV}^T = \boldsymbol{V}^T\boldsymbol{V} = \boldsymbol{I}$ and $\Lambda = \text{Diag}(\lambda)$ is a diagonal matrix of eigenvalues $\lambda$.

$$
\begin{aligned}
x^T \left(\boldsymbol{B}^T \boldsymbol{A} + \boldsymbol{AB}\right) x &= x^T \left(\boldsymbol{B}^T \boldsymbol{V}\Lambda\boldsymbol{V}^T + \boldsymbol{V}\Lambda\boldsymbol{V}^T \boldsymbol{B}\right) x \\
&= x^T \boldsymbol{I}\boldsymbol{B}^T \boldsymbol{V}\Lambda\boldsymbol{V}^T \boldsymbol{I}x + x^T \boldsymbol{I}\boldsymbol{V}\Lambda\boldsymbol{V}^T \boldsymbol{B}\boldsymbol{I}x \\
&= x^T \boldsymbol{VV}^T \boldsymbol{B}^T \boldsymbol{V}\Lambda\boldsymbol{V}^T \boldsymbol{VV}^T x + x^T \boldsymbol{VV}^T \boldsymbol{V}\Lambda\boldsymbol{V}^T \boldsymbol{B}\boldsymbol{VV}^T x \\
&= x^T \boldsymbol{V} \left(\boldsymbol{V}^T \boldsymbol{B}^T \boldsymbol{V}\Lambda + \Lambda\boldsymbol{V}^T \boldsymbol{B}\boldsymbol{V}\right) \boldsymbol{V}^T x \\
&= (\boldsymbol{V}^T x)^T \left(\boldsymbol{V}^T \boldsymbol{B}^T \boldsymbol{V}\Lambda + \Lambda\boldsymbol{V}^T \boldsymbol{B}\boldsymbol{V}\right) (\boldsymbol{V}^T x)
\end{aligned}
$$

Let $\Gamma = \boldsymbol{V}^T \boldsymbol{BV}$ which is equivalent to $\boldsymbol{B} = \boldsymbol{V}\Gamma\boldsymbol{V}^T$, we get that $x^T \left(\boldsymbol{B}^T \boldsymbol{A} + \boldsymbol{AB}\right) x = O(\epsilon)$ for all $x$ with $\|x\|_2 = 1$ is equivalent to $x^T \left(\Gamma^T \Lambda + \Lambda\Gamma\right) x = O(\epsilon)$ for all $x$ with $\|x\|_2 = 1$ (as $\left\|\boldsymbol{V}^T x\right\|_2 = \|x\|_2$).

This is further equivalent to the condition that $\max_{\|x\|_2=1} x^T \left(\Gamma^T \Lambda + \Lambda\Gamma\right) x = \left\|\Gamma^T \Lambda + \Lambda\Gamma\right\|_2 = O(\epsilon)$. Finally, we see that $\|\boldsymbol{M}\|_2 \leq \|\boldsymbol{M}\|_F \leq \sqrt{r} \|\boldsymbol{M}\|_2$ holds true for all $r$-rank matrices $\boldsymbol{M}$. Thus, we have

$$\sum_{ij} (\lambda_j \Gamma_{ji} + \lambda_i \Gamma_{ij})^2 = \left\|\Gamma^T \Lambda + \Lambda\Gamma\right\|_F^2 = O_r(\epsilon^2)$$

$\square$

Considering the result in Theorem A.1, there are many linear transformations $\boldsymbol{L}$ that satisfy equation equation 27. We restrict our attention to the only matrices given by the anstaz motivated by the following corollary.

**Theorem A.2.** *For any matrix $\boldsymbol{B}$, we can divide $\boldsymbol{B}$ into two matrices : symmetric matrix $\boldsymbol{B}_S = \frac{1}{2} \left(\boldsymbol{B} + \boldsymbol{B}^T\right)$ and anti-symmetric matrix $\boldsymbol{B}_A = \frac{1}{2} \left(\boldsymbol{B} - \boldsymbol{B}^T\right)$. We can further restrict $\boldsymbol{B}_S$ and $\boldsymbol{B}_A$ to the following family of matrices to give an ansatz for constructing an $\boldsymbol{B}$ that satisfies equation equation 28*

- *$\boldsymbol{B}_S$ has dominant modes only along the slow modes of the $\boldsymbol{A}$*

- *$\boldsymbol{B}_A$ has a block diagonal structure which mixes almost-degenerate eigenspaces of $\boldsymbol{A}$.*

*Proof.* Let $\Gamma_S = \frac{1}{2}\left(\Gamma + \Gamma^T\right)$ and $\Gamma_A = \frac{1}{2}\left(\Gamma - \Gamma^T\right)$. We start by observing that $\boldsymbol{B}_S = \frac{1}{2}\left(\boldsymbol{B} + \boldsymbol{B}^T\right) = \frac{1}{2}\left(\boldsymbol{V}\Gamma\boldsymbol{V}^T + \boldsymbol{V}\Gamma^T\boldsymbol{V}^T\right) = \boldsymbol{V}\frac{1}{2}\left(\Gamma + \Gamma^T\right)\boldsymbol{V}^T = \boldsymbol{V}\Gamma_S\boldsymbol{V}^T$. Similarly we can show that $\boldsymbol{B}_A = \boldsymbol{V}\Gamma_A\boldsymbol{V}^T$.

$$
\begin{aligned}
\sum_{ij}\left(\lambda_i\Gamma_{ij} + \lambda_j\Gamma_{ji}\right)^2 &= \sum_{ij}\left(\lambda_i\frac{(\Gamma_S)_{ij} + (\Gamma_A)_{ij}}{2} + \lambda_j\frac{(\Gamma_S)_{ij} - (\Gamma_A)_{ij}}{2}\right)^2 \\
&\leq \frac{1}{2}\sum_{ij}\left(\lambda_i + \lambda_j\right)^2(\Gamma_S)_{ij}^2 + \frac{1}{2}\sum_{ij}\left(\lambda_i - \lambda_j\right)^2(\Gamma_A)_{ij}^2
\end{aligned}
$$

Thus it suffices to have

$$
\sum_{ij}\left(\lambda_i + \lambda_j\right)^2(\Gamma_S)_{ij}^2 = O_r(\epsilon^2) \quad \text{and} \quad \sum_{ij}\left(\lambda_i - \lambda_j\right)^2(\Gamma_A)_{ij}^2 = O_r(\epsilon^2) \tag{29}
$$

As a result of the first part, we get that $(\Gamma_S)_{ij} = O(\epsilon)$ if $\max(\lambda_i, \lambda_j) \neq O(\epsilon)$ which shows that the dominant modes of $\boldsymbol{B}_S$ are along the slow modes of $\boldsymbol{A}$. Similarly, the second part gives us that $(\Gamma_A)_{ij} = O(\epsilon)$ if $|\lambda_i - \lambda_j| \neq O(\epsilon)$ which shows that $\boldsymbol{B}_A$ only mixes the almost-degenerate eigenspaces (space of eigenvectors whose the eigenvalues are very close together) of $\boldsymbol{A}$. $\qquad\square$

## B. Experimental Setup : AMBER Force-field

We implement a simplified force-field with implicit solvent (i.e. water molecules are not modeled and appear as hydrogen-bonding and hydrophobicity terms. In protein folding our energy function consists of five potential energies for: bond length $E_{bond}$, bond angles $E_{angle}$, Van der Waals $E_{vdW}$, hydrophobic $E_{hp}$ and hydrogen bonding $E_H$ (Ceci et al., 2007).

**Protein Folding with Classical MD Using AMBER Force Field** We incorporate the AMBER force field, known for its accurate representation of molecular interactions, particularly in proteins. This force field is implemented using the parameters from OpenMM (Eastman et al., 2017), and it comprehensively models the following interactions:

- Bond lengths $E_{bond}$ and bond angles $E_{angle}$

- Torsional angles $E_{torsion}$

- Non-bonded interactions including van der Waals $E_{vdW}$ and electrostatic $E_{elec}$ forces

We utilize the functional forms and parameters specified in the AMBER force field:

$$E_{bond} = \sum_{bonds} k_{bond}(r - r_0)^2 \qquad E_{angle} = \sum_{angles} k_{angle}(\theta - \theta_0)^2 \tag{30}$$

$$E_{torsion} = \sum_{torsions} V_n \left[1 + \cos(n\omega - \gamma)\right] \qquad E_{vdW} = \sum_{i<j} \frac{A_{ij}}{r_{ij}^{12}} - \frac{B_{ij}}{r_{ij}^6} \tag{31}$$

$$E_{elec} = \sum_{i<j} \frac{q_i q_j}{4\pi\epsilon_0\epsilon_r r_{ij}} \tag{32}$$

Here, $r$ and $\theta$ represent the bond lengths and angles, respectively, with $r_0$ and $\theta_0$ as their equilibrium values. The torsional term $E_{torsion}$ includes a sum over all torsion angles $\omega$, with periodicity $n$, amplitude $V_n$, and phase $\gamma$. The Lennard-Jones potential in $E_{vdW}$ is characterized by parameters $A_{ij}$ and $B_{ij}$, and $E_{elec}$ is calculated using the Coulombic potential with partial charges $q_i$, $q_j$ and the relative permittivity $\epsilon_r$.

In this simulation, we exclude the modeling of solvent effects entirely, focusing solely on the protein in vacuum. This approach simplifies the computational model while emphasizing the direct interactions within the protein.

The overall energy of the system is then given by:

$$\mathcal{L}(X) = E_{bond} + E_{angle} + E_{torsion} + E_{vdW} + E_{elec} \tag{33}$$

## C. Further Experimental Results

### C.1. Energy Results

In table 3 and table 4, we list for each simulation method and for each experiment, the value of the lowest energy configuration found structurally close to a given conformer. The closeness of the values to the ground truth found by the long simulation show that the configurations found are valid conformers. As the direct optimization method has two $\epsilon$ parameters (units nanometers), one for sampling around the starting point to estimate the Hessian (eqs 5 and 7), and one for the maximization step to choose best $\mathbf{L}$'s (eq 15), we report two different experiments with varying values of the parameters.

**Vacuum Results.** The last three columns are for the three prominent conformations.

| Method | $C_5$ | $C_7^{ax}$ | $C_7^{eq}$ |
|---|---|---|---|
| Long Simulation | -80 | -77 | -83 |
| Direct Optim $\epsilon = (0.01, 0.1)$ | -80 | -84 | -78 |
| Direct Optim $\epsilon = (0.01, 0.5)$ | -80 | -78 | -78 |
| Full Hessian | -80 | -78 | -84 |
| $\mathbf{H}_2$ backbone (slow subspace) | -80 | -84 | -84 |
| $\mathbf{H}_2$ backbone (degenerate subspace) | -80 | -78 | -84 |

Table 3: Energy($kJ/mol$) in Vacuum for discovered Conformers of Alanine-Dipeptide

**Solvent Results.** These are for simulations in water. The last three columns are for the three prominent conformations. "Solvent Direct Optim" means using the symmetry loss eq 3 and 5 with the energy function of a forcefield that includes explicit solvent terms (using openMM).

| Method | $\alpha_L$ | $\beta$ | $\alpha_R$ |
|---|---|---|---|
| Long Simulation | -120 | -127 | -128 |
| Direct Optim $\epsilon = (0.01, 0.1)$ | -105 | -127 | -128 |
| Direct Optim $\epsilon = (0.01, 0.5)$ | -120 | -127 | -128 |
| Solvent Direct Optim $\epsilon = (0.01, 0.1)$ | -128 | -127 | -120 |
| Solvent Direct Optim $\epsilon = (0.01, 0.5)$ | -97 | -127 | -128 |
| Full Hessian | -120 | -127 | -128 |
| $\mathbf{H}_2$ backbone (slow subspace) | -128 | -127 | -128 |
| $\mathbf{H}_2$ backbone (degenerate subspace) | -120 | -127 | -128 |

Table 4: Energy($kJ/mol$) in Solvent for discovered Conformers of Alanine-Dipeptide

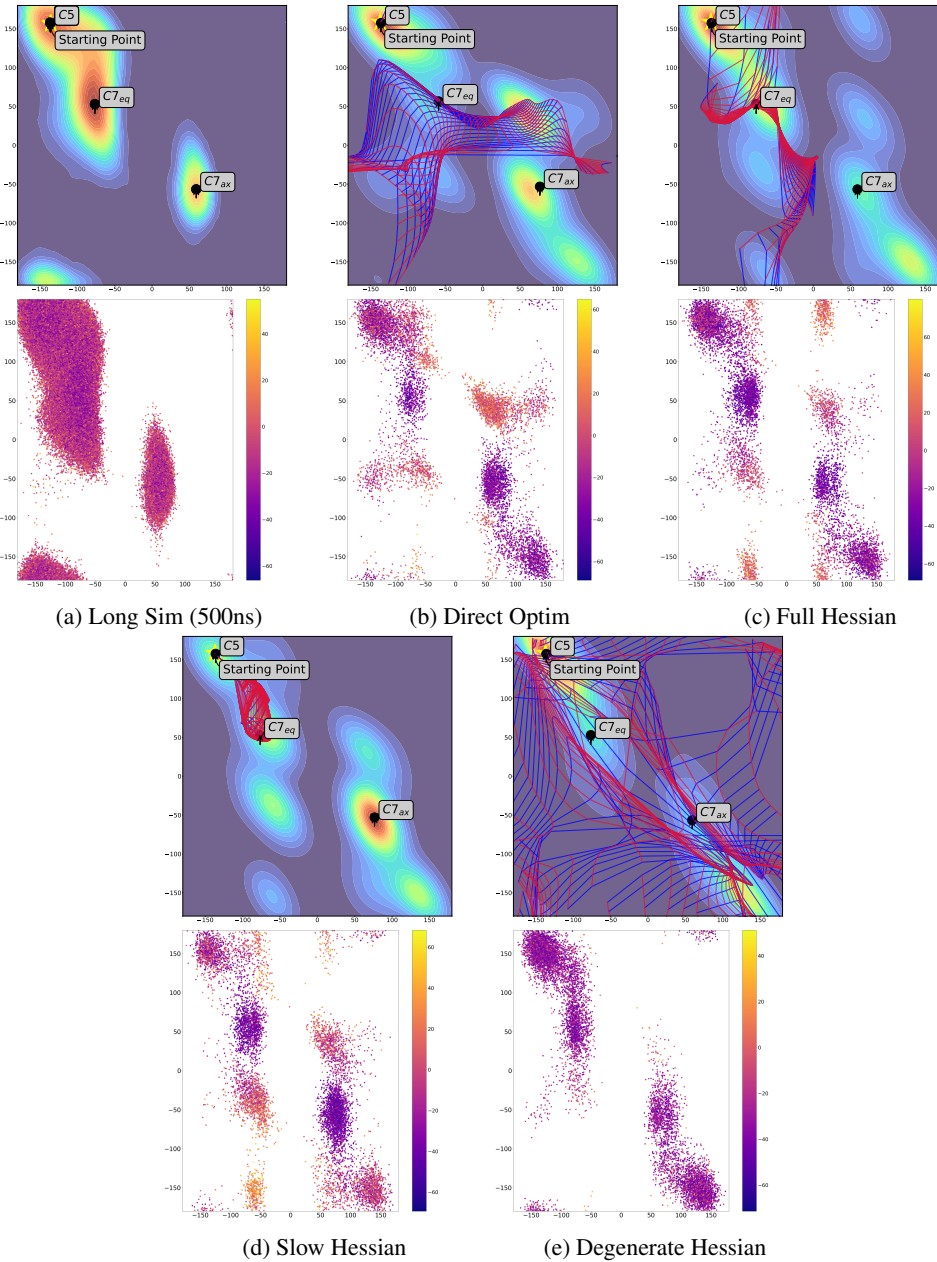

Figure 6: Ramachandran plot for alanine dipeptide in Vacuum based on the a) long (500 ns) openMM simulation with implicit solvent starting at $\beta$ alanine dipeptide, b) direct optimization based approach over the molecular forcefield, c) analytically solving the full $\sigma^4$ term in equation 4 d) slow subspace of $\boldsymbol{H}_2$ e) fast degenerate subspace of $\boldsymbol{H}_2$ f) direct optimization based approach over the solvent and the molecular forcefield . The blue and red grid lines on the plots refer to the grid traced by transforming $\beta$ alanine using the two most effective DOF discovered by our algorithms. Additionally the scatter plot gives the values of the potential energy (in presence of solvent) at the points sampled.

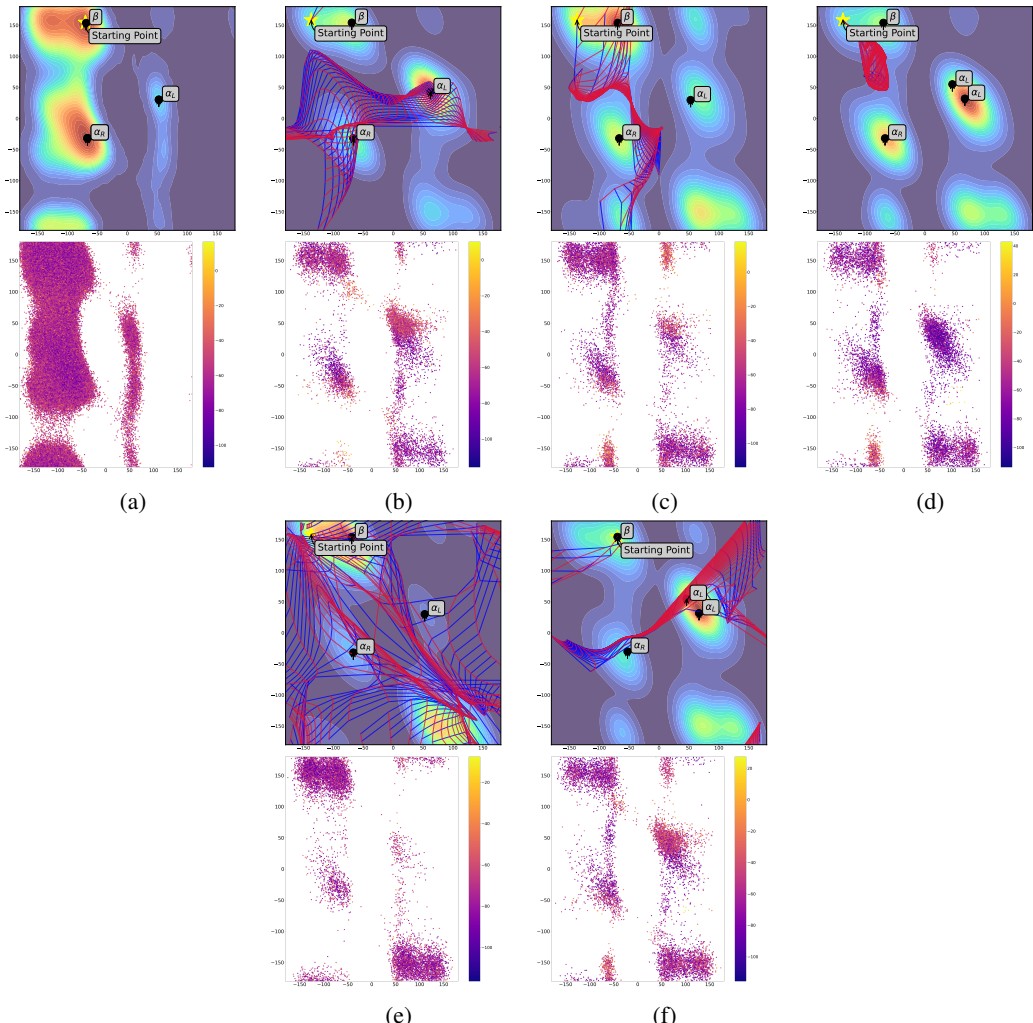

Figure 7: Ramachandran plot for alanine dipeptide in water based on the a) long (500 ns) openMM simulation with implicit solvent starting at $\beta$ alanine dipeptide, b) direct optimization based approach over the molecular forcefield, c) analytically solving the full $\sigma^4$ term in equation 4 d) slow subspace of $\boldsymbol{H}_2$ e) fast degenerate subspace of $\boldsymbol{H}_2$ f) direct optimization based approach over the solvent and the molecular forcefield . The blue and red grid lines on the plots refer to the grid traced by transforming $\beta$ alanine using the two most effective DOF discovered by our algorithms. Additionally the scatter plot gives the values of the potential energy (in presence of solvent) at the points sampled using the corresponding method.

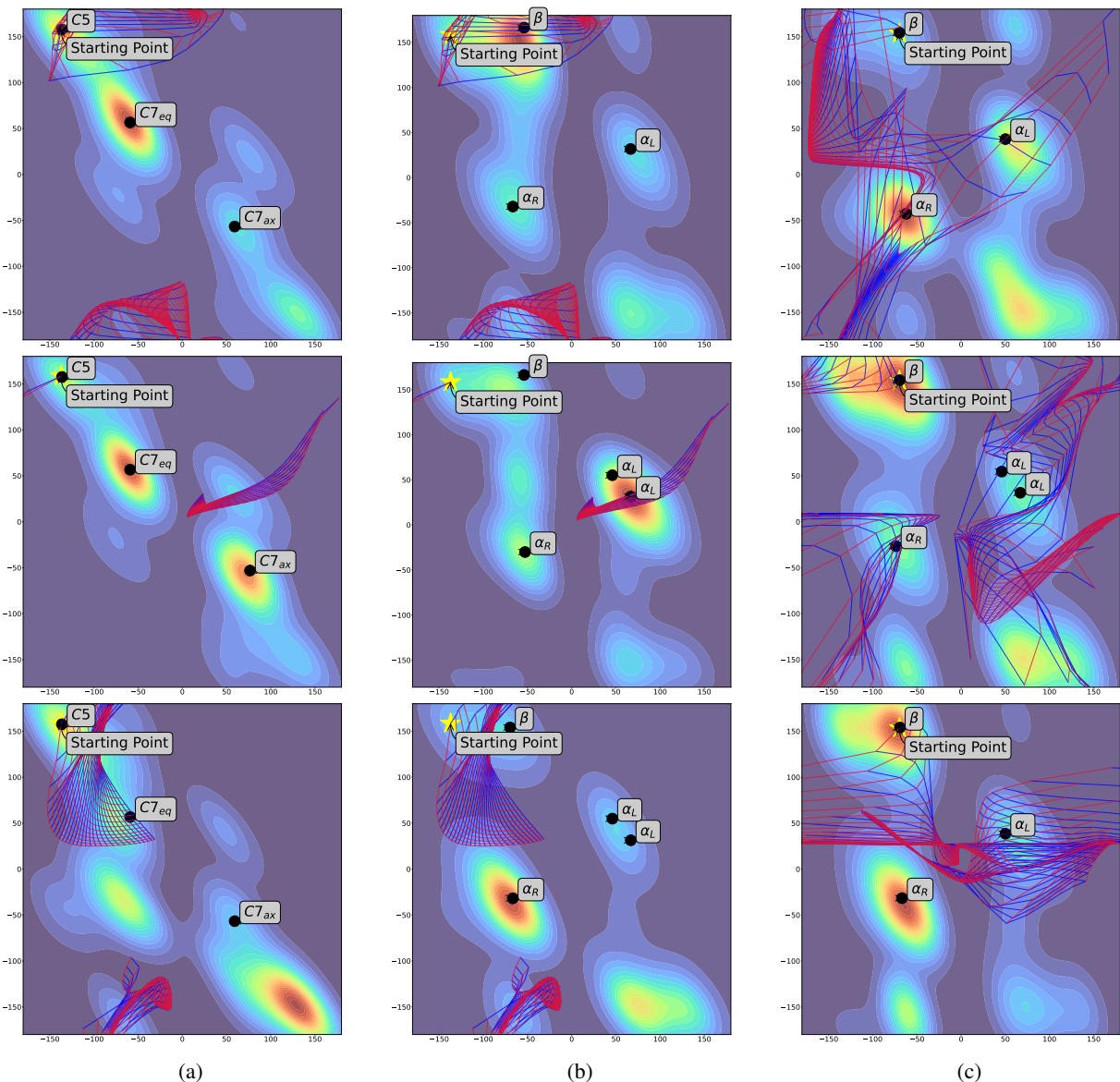

Figure 8: The gridlines and conformers found for three independent runs (row 1, row 2 and row 3) using the direct optimization method with $\varepsilon_1 = 0.1$ and $\varepsilon_2 = 0.01$. a) give the results in vacuum b) gives the results where the simulation is conducted with solvent but the initial trajectory is derived without solvent and c) where the simulation is conducted in solvent and the initial optimization problem is also solved using the solvent forcefield.

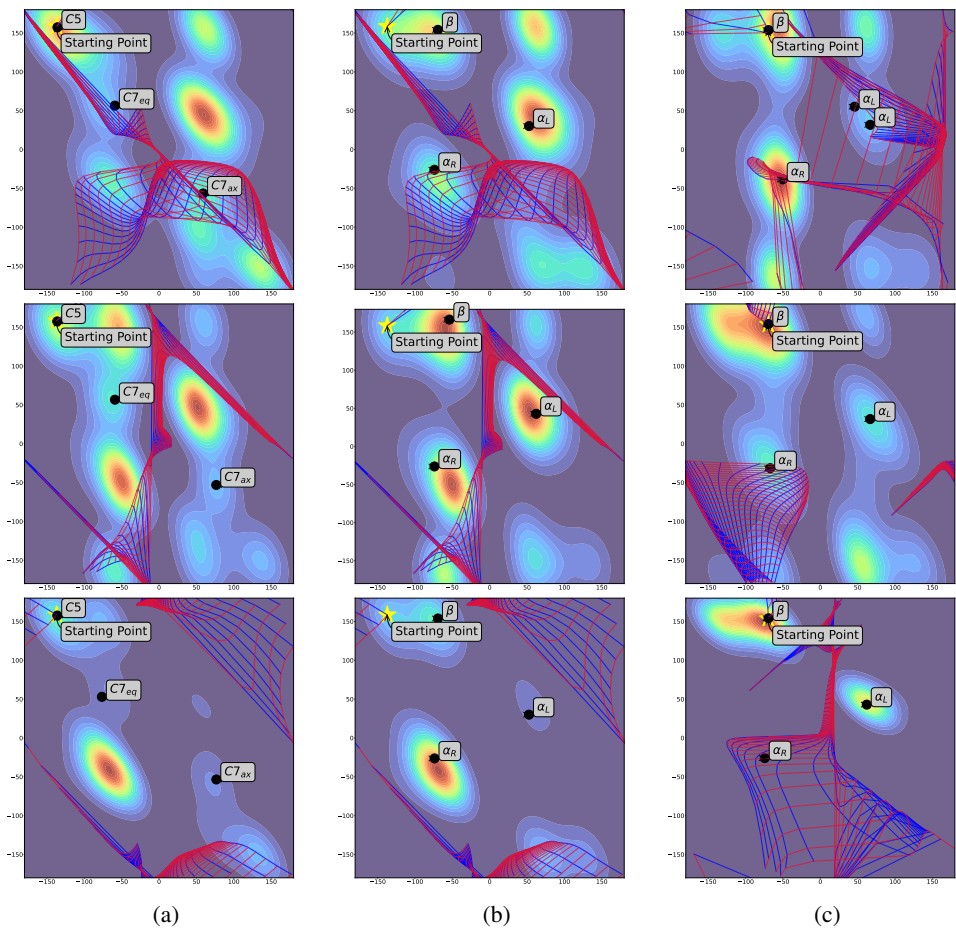

Figure 9: The gridlines and conformers found for three independent runs (row 1, row 2 and row 3) using the direct optimization method with $\varepsilon_1 = 0.5$ and $\varepsilon_2 = 0.01$. a) give the results in vacuum b) gives the results where the simulation is conducted with solvent but the initial trajectory is derived without solvent and c) where the simulation is conducted in solvent and the initial optimization problem is also solved using the solvent forcefield.

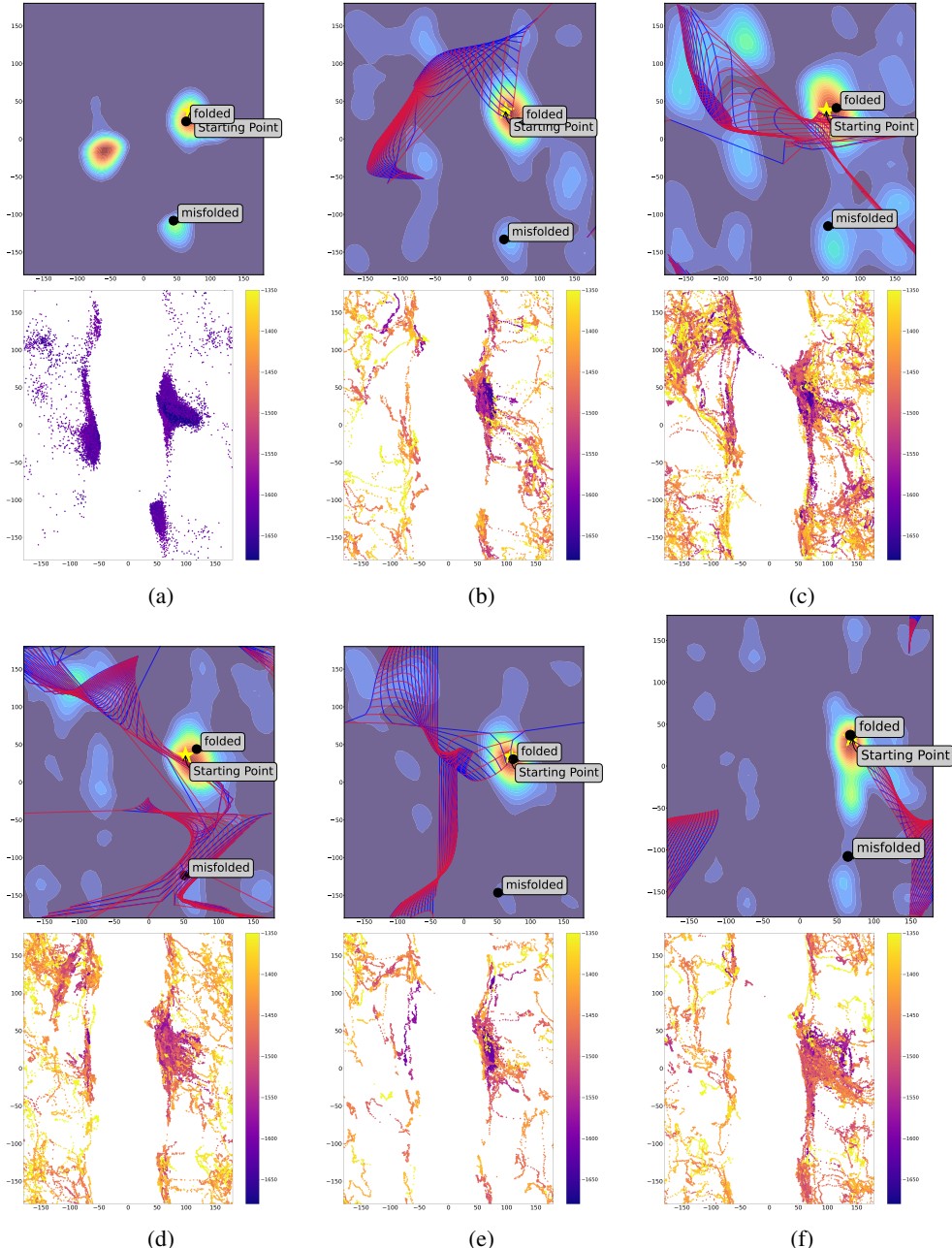

Figure 10: Ramachandran plot for alanine dipeptide in water based on the a) long (500 ns) openMM simulation with implicit solvent starting at folded chignolin, b) direct optimization based approach over the molecular forcefield with symmetry discovered at close range (0.1 nm), c) direct optimization based approach over the molecular forcefield with symmetry discovered at close range (0.1 nm), d) analytically solving the full $\sigma^4$ term in equation 4 e) direct optimization based approach over the solvent and the molecular forcefield with symmetry discovered at long range (0.1 nm f) direct optimization based approach over the solvent and the molecular forcefield with symmetry discovered at long range (0.5 nm). The blue and red grid lines on the plots refer to the grid traced by transforming folded chignolin using the two most effective DOF discovered by our algorithms.

