# OpenReview forum: "Symmetry-Driven Discovery of Dynamical Variables in Molecular Simulations"
_ICML.cc/2025/Conference — ICML 2025 poster_

### Official Review · Reviewer_YyW7 · 2025-03-12

**Overall Recommendation:** 1

**Summary:**

The paper introduces a framework for discovering effective degrees of freedom (DOF) in molecular simulations by identifying approximate symmetries of the energy function. Instead of relying on simulation trajectories or training datasets, the authors formulate an optimization problem—called a “symmetry loss”—and, in one approach, use second-order (Hessian-based) information to find transformations that keep the energy nearly invariant. These transformations become a set of collective variables or DOF that help explore low-energy regions of the molecular landscape more efficiently. The authors demonstrate their methods on alanine dipeptide (in both vacuum and implicit solvent) and on chignolin, showing that the discovered DOF can reach known conformers and reveal states that are traditionally harder to sample with basic molecular dynamics. Overall, the paper’s main contribution is a data-free technique that systematically derives low-energy directions from the force field itself, offering a way to speed up or enrich sampling of molecular conformations.

**Claims And Evidence:**

The paper’s key theoretical claim—namely, that low-energy degrees of freedom can be discovered by finding approximate symmetries of the energy landscape—receives strong support through detailed derivations and proofs. The authors demonstrate mathematically how to use the gradient or Hessian of the force field to identify transformations that leave the energy almost unchanged, then validate these ideas on small systems (alanine dipeptide and chignolin). This evidence is clear and consistent for the claim that the proposed methods can uncover physically relevant low-energy directions in the configuration space.

However, the assertion that these methods lead to significantly more diverse sampling or noticeable speedups compared to established enhanced-sampling techniques is less rigorously substantiated. The authors do present examples where their approach discovers states that standard molecular dynamics simulations take longer to find, but there is no in-depth, head-to-head comparison of sampling efficiency (e.g., run-time or number of force evaluations) against commonly used methods. Thus, while the theoretical foundation and small-scale results strongly support the symmetry-based framework, the more expansive claims about large gains in sampling diversity or efficiency remain only partially demonstrated.

**Essential References Not Discussed:**

All the relevant references are discussed.

**Experimental Designs Or Analyses:**

The experimental approach—focusing on alanine dipeptide (both vacuum and implicit solvent) and chignolin (implicit solvent)—is broadly sound as a preliminary demonstration that the discovered transformations can access known conformers. Both systems are widely recognized testbeds for validating the discovery of local minima and dihedral-angle variations. A long MD simulation serves as a reference to confirm that the proposed method recovers states consistent with standard MD. The procedure involves extracting approximate symmetries at a reference configuration, systematically applying transformations (on a grid) to generate new structures, and then minimizing and running short simulations to confirm low-energy stability. This is appropriate to gauge whether the method meaningfully covers conformational diversity; however, the study does not provide comparisons to established sampling methods. Moreover, while the results are promising, the scope is limited to relatively small systems, leaving open questions on scalability, performance with explicit solvent, and how comprehensively the new approach samples large, complex conformational landscapes.

**Methods And Evaluation Criteria:**

The authors focus on two well-known test systems—alanine dipeptide and chignolin—both of which are standard benchmarks in molecular dynamics. These systems provide clear baselines for verifying whether new approaches capture known conformers and relevant collective variables. Because the paper’s main claim is that approximate symmetries of the energy can yield physically meaningful DOF, using these relatively small yet widely studied molecules is a sensible choice: it makes it straightforward to check whether the discovered degrees of freedom align with known dihedral angles or populated states. That said, the evaluation consists mainly of seeing whether the method recovers recognized conformations and can occasionally reach less-populated states; it does not include more comprehensive comparisons against standard enhanced-sampling approaches.

**Other Comments Or Suggestions:**

I don’t think that this work lies in the field of machine learning, so I would recommend that the authors submit it to some applied biology conferences or journals.

I see potential in this paper, but it requires significant refinement—especially in how the methodology is presented and how it is compared with competing approaches.

**Other Strengths And Weaknesses:**

List of typos:
Line 34, «approximate»
Line 131, missed space in front of «of»
Line 275, «assume», «meaning»
Line 288, «only»
Line 316, «minimizing»
Line 551, 569, «eq 5» and similar — no reference
Line 681, «DOF»
Line 715, «One»
Line 718, has to be «degenerate eigenspaces for each»

**Questions For Authors:**

No questions

**Relation To Broader Scientific Literature:**

The paper addresses the longstanding problem of identifying collective variables for molecular simulations, a task also tackled by methods such as metadynamics, replica-exchange MD, and various data-driven approaches. Unlike those methods, which often rely on extensive trajectories or predefined biases, this work derives low-energy directions straight from the force field’s gradient and Hessian, placing it closer in spirit to normal-mode analyses used in protein dynamics.

**Theoretical Claims:**

The paper’s main derivations—specifically, using Taylor expansions around a reference configuration and linking approximate energy invariance to low-curvature or degenerate directions of the Hessian—align with standard linear algebra and perturbation results. The statement that degenerate eigenvalues in the Hessian induce rotation-like symmetries follows well-known principles. While minor indexing or notation issues are present, they do not invalidate the arguments, and the proofs appear consistent with conventional theoretical foundations.

---

> ### Author Rebuttal · Authors · 2025-04-01
>
> We thank you for your review and we hope we can address some of your concerns below :
>
> ### Quantitative Metrics and Baselines ###
> We understand your concerns regarding the rigor of the evaluation criteria. Please refer to the response for Reviewer 9HNJ to view a table of quantitative results that might address some of the concerns you raised. As seen in the table, the grid of initial configurations given by the discovered DoF provides coverage of the entire conformational landscape. As a result, we are able to find at least some points within each conformational basin within only 10 simulation steps.
>
> To address your second point about the lack of comparison with other enhanced sampling methods, we find other method to be orthogonal in nature to some of the other sampling methods available. Most enhanced sampling methods like umbrella sampling and metadynamics based methods require specified CVs in order to sample the space. As a result these methods, work on a reduced low-dimensional space unlike the our proposed method which directly works on the all atom model. While replica exchange methods do directly sample from the high-dimensional all-atom positions, they do not involve learning any local symmetries. Furthermore the replica exchange methods require communication between different replicas and cannot explore different parts of the energy landscape simultaneously. We provide a highly parallelizable method which uses local symmetry information to efficiently index and explore conformational basins. Using parallel exploration we only need to simulate <10 steps/starting configuration (total 1000 starting configuration) to get to all the known conformational basins for the explored systems. As a result, the method cannot be directly compared to the sequential algorithms like metadynamics and replica exchange models.
>
> ### Scaling to Larger Systems ###
>
> As mentioned in the paper, the direct optimization based methods can be applied to any system immersed in a force field. Even in the presence of explicit solvent, the procedure only needs to access the force exerted on the atoms in the molecule for given configurations. As correctly stated by the reviewer, the paper is introducing the idea of discovering local DoF, we tried to focus mostly on small systems with well-known and interpretable DoF.
>
> ### Concerns Regarding Applicability to ML ###
>
> We believe the symmetry discovery/optimization part of the procedure as well as the problem of transfer and generalizability of learnt DoF is an ML algorithm, an ML venue is much better qualified to judge its validity than a bio venue. Additionally, comp bio, drug discovery and physical simulation are among the fastest growing and most important use cases of ML.

---

> > ### Comment · Reviewer_YyW7 · 2025-04-06
> >
> > I see potential in this paper, but it requires significant refinement—especially in how the methodology is presented and how it is compared with competing approaches. While I understand that methods such as metadynamics or replica exchange may not be directly comparable, it would be valuable for the paper to include a clear discussion of these methods’ limitations and a comparison that accounts for those constraints. Without such an analysis, it is hard to fully assess the proposed method, and thus I am not inclined to change my current evaluation.

---

> > > ### Author Response · Authors · 2025-04-07
> > >
> > > We thank the reviewer for their appreciation of the potential of the paper.
> > >
> > > ### Improvement in Presentation ###
> > >
> > > We have provided some points towards improving the presentation in our responses to reviewer p5r2. We feel the additional glossary of notation and a section illustrating the use of the proposed method on a small sample system would significantly improve the presentation. Please check our rebuttal to reviewer p5r2 and the rebuttal response to reviewer zCrM for an example illustrating the use of our proposed methods for symmetry discovery. We feel this example should demonstrate the key ideas of the paper.
> > >
> > > ### Comparision to Enhanced Sampling Methods ###
> > >
> > > We understand the reviewer's suggestion to compare our methods to existing enhanced sampling methods. As noted in our earlier response, we would like to highlight that CV-based methods like Umbrella Sampling and metadynamics are known to be capable of exploring only low-dimensional spaces spanned by CVs, but our current method tries to explore the high-dimensional space spanned by the all-atom positions. Thus, the only enhanced sampling method that is somewhat comparable is the family of Replica Exchange Molecular Dynamics (REMD) methods. We can add a more in-depth discussion regarding these methods. As suggested by the reviewer, we add the REMD simulation baseline results here for alanine-dipeptide to contrast it with the results we have for our proposed methods (given in our rebuttal to reviewer 9HNJ). Here we use REMD simulation with 8 parallel simulations at logarithmically spaced out temperatures between 300 K - 500 K with an attempted transition between adjacent temperatures every 50 steps ($0.1 ps$).
> > >
> > > |                             |        | 2 ps | 10 ps | 20 ps | 0.1 ns | 0.2 ns | 1 ns   | 2 ns   | 10 ns  | 20 ns  |
> > > | --------------------------- | ------ | -------- | ------- | ------- | ------ | ------ | ------ | ------ | ------ | ------ |
> > > |  | $C5$  | 0.4659   | 0.3676  | 0.2463  | 0.2168 | 0.2264 | 0.2082 | 0.2232 | 0.2340 | 0.2316 |
> > > | Alanine Dipeptide in Vacuum | $C7_{eq}$ | 0.1818 | 0.2745   | 0.3515  | 0.3745  | 0.3708 | 0.3311 | 0.3357 | 0.3367 | 0.3302 |
> > > | |$C7_{ax}$ | 0.0000 | 0.0000   | 0.0000  | 0.0000  | 0.0000 | 0.0768 | 0.0467 | 0.0226 | 0.0332 |
> > > |                              |          |          |         |         |        |        |        |        |        |        |
> > > | | $\\beta$ | 0.9205   | 0.7451  | 0.6052  | 0.4808 | 0.4336 | 0.4359 | 0.4422 | 0.4505 | 0.4557 |
> > > |  Alanine Dipeptide in Solvent | $\\alpha_L$ | 0.0000   | 0.0000   | 0.0000  | 0.0000  | 0.0000 | 0.0017 | 0.0063 | 0.0061 | 0.0065 |
> > > | |$\\alpha_R$ | 0.0000   | 0.0956   | 0.2166  | 0.3079  | 0.3417 | 0.3383 | 0.3238 | 0.3200 | 0.3153 |
> > >
> > > We use smaller timescales here than the timescales in the long simulation baseline, as the convergence is faster. However, we still see that the proposed methods are faster by an order of magnitude in terms of simulation time (our proposed methods find all conformations within $20 fs$ of simulation time). Moreover, we also note that unlike REMD, which requires some interaction (exchanging) between the parallel simulations being run on different temperature scales, the parallel simulations used in our method are fully independent. This makes the simulations given by our method even faster in terms of wall-clock time.
> > >
> > > We hope these additional results address the concerns raised by the reviewer.

---

### Official Review · Reviewer_zCrM · 2025-03-12

**Overall Recommendation:** 3

**Summary:**

This paper proposes a method to identify effective degrees of freedom (DOF) of dynamics by connecting them with approximate symmetries of the energy function. This is done through identifying a state $x$ with a group element (here restricted to the general linear group) whose action on $x_0$ (a reference state) produces $x$. For sufficiently close $x$ to $x_0$, this can then be parameterised by a linear combination of basis vectors of the Lie algebra. These symmetry based DOFs are optimised according to two principles: the transformations should not cause large changes in energy globally (so that main conformations are still distinct under this parameterisation), but should cause enough change in energy to overcome local energy barriers. This results in loss functions that can either be directly optimised, or under a local assumption be used to derive analytical loss expressions involving the Hessian. Both approaches are tested on two relatively simple test problems (alanine dipeptide and Chignolin), where it is shown that these eDOFs are useful in exploring the conformation landscape.

**Claims And Evidence:**

The two examples presented demonstrate to a certain extent the effectiveness of the method, although some quantitative evaluation of the effectiveness of the DOFs would have been better. Moreover, there lacks a general comparison with other methods which tries to construct eDOFs (such as collective variables), although such a comparison may be difficult without a well-defined evaluation criterion. At the very least, one could compare meta-dynamics driven by learned CVs and the current method on the Ramachandran plot. Note that meta-dynamics can be used to perform sampling from the correct distribution, and compute the free energy, but the current method does not seem to be able to do this. The authors may wish to comment on this, and perform some numerical comparison.

**Essential References Not Discussed:**

It may be useful to discuss https://arxiv.org/pdf/2307.00365 and methods therein for learning/constructing collective variables.

**Experimental Designs Or Analyses:**

The experiments are reasonably designed, although as discussed above the evaluation and ablation can be improved.

**Methods And Evaluation Criteria:**

The visualisation of the Ramachandran plot is useful in demonstrating that the eDOF constructed can be used to probe the conformations, although I do not believe it is easy to perform quantitative comparisons with different methods.

There is also a lack of ablation studies. For example, can one compare the Ramachandran plots of the case which carries out steps 1-5 on page 7, except in step 2 one uses either 1) two randomly sampled $L_1, L_2$; or 2) replaces $L_1 x_0$ and $L_2 x_0$ by the top two eigenvectors of the Hessian of $E$ at $x_0$?

**Other Comments Or Suggestions:**

- Abstract: "... we do not require data and rely on knowledge ..." -> "... we do not require data but rely on knowledge ..."

- Page 1: "we don’t want to change in energy to be so high that it would break chemical bonds" -> Grammar issue

- Page 1: "both computationally efficient and physically insightful." -> what do you mean by physically insightful?

- Page 3: I find the notation for general linear group strange. It is usually $\text{GL}(n)$ or  $\text{GL}(n,F)$ if you want to specify the field, but $\text{GL}(\mathcal X)$ doesn't make sense to me, and moreover it is not consistent with the subsequent usage of (special) Euclidean groups (which are also not properly defined in the text).

- Page 3: The Orbit defintion seems trivial as written, since its either $0$ or whole space minus 0. Do you mean to consider a subgroup?

- Page 3: The symbols used in the section under "Group parameters as DOF" is quite unclear to me.

  - For the notation of the Lie algebra basis $\mathbf{L}_a$, is this a single basis element, or does $a$ run through the dimension of the Lie algebra?
  - For the notation $\theta \cdot \mathbf{L}$, does this mean $\sum_a \theta_a \mathbf{L}_a$ or something else? It is not defined.
  - If $\theta$ is a vector, the Taylor expansion error term should be $O(|\theta|^2)$.
  - Since it appears to me that we are always considering the general linear group, what's the point of this abstraction to Lie groups/algebras? Why not just identify the Lie algebra with $M_n(\mathbb R)$?
  - Boldfaced $\mathbf{I}$  is used but subsequently $I$ is used instead (e.g. page 4). Is there any meaning to the different notations?

  Overall, I suggest the authors properly define the notations (even if it is placed in the appendix). These issues affect readability of the core parts of the paper.

- Page 4: "set of DOF at by" -> Grammar issue

- Page 4: Equation 1 is not clear, is $S$ considered a subset of $GL(n,\mathbb R)$ or $M_n(\mathbb R)$? It should be the latter given equation 2. Also, what's the dependence of $\delta E$ on DOF? Why not just use $E_\text{barrier}$ which is already used in the previous page and not introduce an extra $\eta$?

- Page 4: the variables $\theta_l,\theta_s$ do not seem to appear in equations 1 and 2.  Moreover, subsequent line with $g\approx I + \theta \mathbf{L}$ now takes $\theta$ as a number instead of a vector (as introduced on the previous page).

- Page 5: "where we wanted $\delta E$" -> "where we wanted $\delta E$ to be small"

- Page 6: "menaing" -> Spelling, "oonly" -> Spelling

- Page 6: Equation 15, should $K$ be $n_L$ as in equation (14)?

- Page 6: "minimas" -> "minima"

- Page 6: "presense" -> "presence"

- Page 7: "to find the conformers is settings with" -> Grammar issue

**Other Strengths And Weaknesses:**

trengths

- The problem of finding effective ways (DOFs) to navigate conformational landscape of macromolecules is an important application area, both for qualitative analysis and quantitative computation (sampling, free energy computation, etc)
- The method appears quite interesting and has new ideas that connect (approximate) symmetries of the energy function to effective DOFs.
- The qualitative experiments over two simple but representative problems demonstrate the potential of the method

Weaknesses

- Lack of quantitive evaluations of their method and comparisons with related methods (e.g. learned/constructed CVs as effective DOFs)
- Lack of ablation studies
- Last but not least, while I do like the new ideas in the paper, the presentation of the paper can be *substantially* improved, there are many typos and many expressions lacking proper definition (details below in "Other Comments or Suggestions")

**Questions For Authors:**

1. Can the authors give a minimal example to illustrate your approach and why the DOFs parameterised through symmetries make sense? For example, we can take d=1, n=2 and consider some potential like $E(x_1,x_2) = F(x_1-x_2)+\epsilon\sin(x_2/\epsilon)$ where $F$ is say a double-well potential (or something to that extent, involving potentials of different scales) . We know that the effective DOF should be $x_1-x_2$. It would be helpful to illustrate that the method correctly identifies this.
2. As I understand, since you always take a linear approximation of the exponential map of the general linear group, this is the same as finding perturbation directions as $x_0 \mapsto x_0 + \sum_{i} c_i L_i x_0$. This locally yields "collective variables" in the form of $z_i = L_i x_0$, and if one can piece this together for neighbourhoods of each $x_0$, then one should arrive at some global collective variable? Is this understanding correct? If so, there should be some discussion on this point and the connection of this approach to e.g. the collective variable discovery approaches outlined in https://arxiv.org/pdf/2307.00365.

**Relation To Broader Scientific Literature:**

The related work is generally well written and relevant.

**Theoretical Claims:**

I checked the overall claims and they appear correct. I did not check every step in the derivation in the appendix.

---

> ### Author Rebuttal · Authors · 2025-04-01
>
> Thank you for your thorough and thoughtful review. We address some points below:
>
> ### Quantitative Metrics & Baselines ###
> Please refer to our response to Reviewer 9HNJ for additional results. Although there are existing methods for CV discovery, our method considers the slightly orthogonal task of discovering local DoF. Unlike CVs, the discovered DoF are local in nature and do not necessarily correspond to any global Collective Variable. Moreover, we show that we can sample all the conformers for these molecules within 20 fs (10 steps) of starting the simulation from the sampled gridpoints. This setup is very different from the standard CV setup in metadynamics or replica exchange where the initial configuration is fixed.
> ### Presentation ###
> Thank you for the pointing out the notational inconsistencies. We will add a glossary of the relevant notation in the supplementary material. Furthermore, we will also add a section on illustrative examples applying the proposed method on small systems. Please refer to our response to Reviewer p5r2 for a discussion of a small synthetic system similar to the one you proposed. As the Hessian based method in
> ### Ablation Studies ###
> We will add the suggested ablation studies in the supplementary material. It is not always possible to plot the trajectories for the highest valued eigenvectors of the Hessian as the transformations can lead to unnatural states that lead to errors in the openmm simulation. However, we will add a more detailed discussion in the paper.
> ### Local CV and Relation to other CV discovery work ###
> We thank the reviewer for the reference [1], but the CVs discussed do not align very well with the present work. Although one could define local variables $z_i = L_i x_0$, these resulting variables behave differently from collective variables. We treat all configurations that can be reached from $x_0$ as an equivalence class; thus, all the z_i's would form a basis for this subspace. Thus, all the local $z_i$'s can be used to index the local conformation basin, but it is unclear if these can be patched together or if patching these would give a good global CV.
> ### Clarifications for Typos ###
> - Page 1: "both computationally efficient and physically insightful." -> what do you mean by physically insightful?
> We mean it provides insight into the effective dynamics of the system.
> - Page 3: I find the notation for general linear group strange. … $GL(\mathcal{X})$ doesn't make sense to me, … not consistent with the subsequent usage of (special) Euclidean groups.
> Actually, $GL(V)$ is the standard way the general linear group for a vector space $V$ is denoted, though it may not look as familiar as $GL(n,R)$. We will explain and add the definition of $SE(n)$.
> - Page 3: The Orbit definition seems trivial as written … Do you mean to consider a subgroup?
> You are correct for $\mathcal{X} \sim R^n$. We are indeed usually concerned with a subgroup (the approximate symmetries of $E$). Will reword it.
> - Page 3: symbols in "Group parameters as DOF" is quite unclear to me.
>   - Lie algebra basis $L_a$, is this a single basis element, or does $a$ run through … the Lie algebra?
> $L_a$ is a single basis element.
>   - For the notation $\theta \cdot L$:
> Yes, it is $\sum_a \theta_a L_a$
>   - If θ is a vector, the Taylor expansion error term should be O(|θ|2)
> Any combination $\theta_a \theta_b$ is second order, but, yes, one can pull out the norm of $\theta = |\theta| \hat{\theta}$ and write the error as $O(\theta^2)$.
>   - Since … always GL, what's the point of this abstraction to Lie groups/algebras? Why not just identify the Lie algebra with $M_n(R)$?
> They are matrices, yes, but we want a subgroup
>   - Boldfaced $I$ vs normal. Any difference? No, will fix it.
>   - Overall, I suggest … properly define the notation. Thank you, will do.
> - Page 4: Equation 1 is not clear, is $S$ considered a subset of $GL(n,R)$ or $M_n(R)$? It should be the latter…
> We should clarify in the paper that we are working with the canonical representation of $GL(n,R)$ which maps elements to a subset of $M_n(R)$. So, yes, all in $M_n(R)$.
> Also, what's the dependence of δE on DOF? Why not just use $E_{barrier}$ which is already used in the previous page and not introduce an extra η?
> Eq 3, $\delta E = \theta \nabla E \cdot Lx$ (here $\theta  \in R$) shows dependence of $\delta E$ on DOF $L$. We can replace $\eta$ with $E_{barrier} in eq 1.
> - Page 4: the variables $\theta_l,\theta_s$ do not appear in eqs 1 and 2.
> Sorry, we changed it to $\epsilon_l, \epsilon_s$, will fix.
> - g≈I+θL now takes θ as a number instead of a vector.
> A bit of abuse of notation here, will fix. Here $L$ denotes an arbitrary vector in the Lie algebra instead of specific basis elements $L_a$. Then, $\theta \in R$ is a small $\epsilon$ making clear that $g$ is near identity.
> - P6: Eq 15, $K= n_L$ as in eq (14)?
> Yes, will fix.
>
> [1] Understanding recent deep-learning techniques for identifying collective variables of molecular dynamics

---

> > ### Comment · Reviewer_zCrM · 2025-04-02
> >
> > I think the paper has some nice ideas and I have increased my score assuming that the authors can fix all the notational issues and inconsistencies.
> >
> > Also, the simple example system to demonstrate your approach would be very beneficial. It would be nice to describe it in a reply.

---

> > > ### Author Response · Authors · 2025-04-06
> > >
> > > Thank you for your suggestions. The simple example system is quite interesting to explore. Because of space constraints, we only discuss one possible solution to the minimization problem for two methods instead of exploring the complete solutions. We refer you to our response to Reviewer p5r2 for the setup and part of the solution using the Hessian-based approach. We explore symmetries around given minima $x'$.
> > >
> > > ### Hessian-Based Method ###
> > >
> > > One correction; the hessian should have $\sin(x'_2/\epsilon)$ instead of $\cos(x'_2/\epsilon)$, yielding :
> > > $$ H_E(x') = H_F(|x'_1| - x'_2) \begin{bmatrix} 1 & -\textbf{sgn}(x'_1) \\\\  -\textbf{sgn}(x'_1)& 1 \end{bmatrix} - \frac{\sin(x'_2/\epsilon)}{\epsilon} \begin{bmatrix} 0 & 0 \\\\ 0 & 1 \end{bmatrix} = H_F(|x'_1| - x'_2) \begin{bmatrix} 1 & -\textbf{sgn}(x'_1) \\\\  -\textbf{sgn}(x'_1)& 1 - \frac{-1}{\epsilon H_F(|x'_1| - x'_2)}\end{bmatrix}  = C \begin{bmatrix} 1 & -\textbf{sgn}(x'_1) \\\\  -\textbf{sgn}(x'_1)& \gamma \end{bmatrix}$$
> > > Minima condition gives $\sin$ as $-1$. We refer to $H_F(|x'_1| - x'_2)$ as $\rho$ for brevity. In the previous reply, we explored the case when $\gamma \approx 1$, which corresponds to $\epsilon >> \rho^{-1}$. For $\epsilon << \rho^{-1}$, we have $\gamma \rightarrow \infty$, or rather the approximation $ H_E(x') \approx C \gamma \begin{bmatrix} \gamma^{-1} & -\gamma^{-1}\textbf{sgn}(x'_1) \\\\  -\gamma^{-1}\textbf{sgn}(x'_1)& 1 \end{bmatrix}$
> > > where $C\gamma = \frac{1}{\epsilon}$. Thus, we see that taking $L^* = \begin{bmatrix} 1 & 0 \\\\ 0& 0 \end{bmatrix}$ would give $K_S = a \begin{bmatrix} \gamma^{-1} & -\frac{1}{2}\gamma^{-1}\textbf{sgn}(x'_1) \\\\  -\frac{1}{2}\gamma^{-1}\textbf{sgn}(x'_1)& 0 \end{bmatrix}$. Ignoring the constant $a$, $2 tr(K_S^2) + tr(K_S)^2 = 2 (\gamma^{-2} + \gamma^{-2}\frac{1}{4} + \gamma^{-2}\frac{1}{4} ) + \gamma^{-2} =4 \gamma^{-2} \approx 0$. Thus, $e^{\eta L^*} x = x + (e^\eta - 1)L^* x$ which only changes $x_1$, conserving $x_2 = (e^{\eta L^*} x)_2$.
> > >
> > > The Hessian-based method gives the symmetry present at the lowest scale. If $\epsilon$ is small, we get the symmetry $L^* = \begin{bmatrix} 1 & 0 \\\\ 0& 0 \end{bmatrix}$ $\Big( x_2 = (e^{\eta L^*} x)_2$ corresponding to $x_2$ being conserved $\Big)$. For large $\epsilon$, $L^* = \begin{bmatrix} 1 & \textbf{sgn}(x'_1) \\\\ \textbf{sgn}(x'_1) & 1 \end{bmatrix} \Big( x_1 - \textbf{sgn}(x'_1)x_2 = (e^{\eta L^*} x)_1 - \textbf{sgn}(x'_1)(e^{\eta L^*} x)_2$ is conserved$\Big)$ .
> > >
> > > ### Direct Optimization ###
> > >
> > > For direct optimization method, the infinite sample limit optimization problem can be restated as
> > > $$ \mathbb{E}_{x \sim \mathcal{N}(x', \sigma I)} \left[(\nabla E(x)^\top L x)^2 \right] =  \mathbb{E} \left[tr (x\nabla E(x)^\top L)^2 \right] = \mathbb{E} \left[vec(L)^\top vec (x\nabla E(x)^\top) vec (x\nabla E(x)^\top)^\top vec(L) \right] = vec(L)^\top \mathbb{E} \left[vec (x\nabla E(x)^\top) vec (x\nabla E(x)^\top)^\top\right]  vec(L)  $$
> > >
> > > Assuming a quadratic Hessian-based approximation of $F$ and assuming $\textbf{sgn}(x_1) = \textbf{sgn}(x'_1)$, i.e., minima $x'$ is far from origin we get the following matrix
> > >
> > > $$ H_{E, \sigma} = \begin{bmatrix} A & -\textbf{sgn}(x'_1) A \\\\ -\textbf{sgn} (x'_1) A & A \end{bmatrix} + \begin{bmatrix} 0 & -\textbf{sgn} (x'_1) B \\\\ \textbf{sgn}(x'_1) B & C - 2B \end{bmatrix} $$
> > > where $A$ scales with multiplier $\sigma^2 H_F(|x'_1| - x'_2)^2$, $B$ scales with multiplier $\sigma H_F(|x'_1| - x'_2) \frac{\sigma}{\epsilon} e^{-\left(\frac{\sigma}{\epsilon}\right)^2}$ and $C$ scales with multiplier $1 - e^{-\left(\frac{\sigma}{\epsilon}\right)^2}$. Henceforth, we refer to $H_F(|x'_1| - x'_2)$ as $\rho$.
> > >
> > > Case $\epsilon << \rho^{-1} :$ , for $\epsilon << \sigma << \rho^{-1}$, we have $H_{E, \sigma} \approx \begin{bmatrix} 0 & 0 \\\\ 0 & C \end{bmatrix}$ which yields $L^* = \begin{bmatrix} 1 & 0 \\\\ 0& 0 \end{bmatrix}$.
> > >
> > > However for $ \epsilon << \rho^{-1} << \sigma $, we get $H_{E, \sigma} \approx \sigma^2H_F(|x'_1| - x'_2)^2 \begin{bmatrix} A & -\textbf{sgn}(x'_1) A \\\\ -\textbf{sgn} (x'_1) A & A \end{bmatrix} + \begin{bmatrix} 0 & 0 \\\\ 0 & C \end{bmatrix} $ which yields $L^* = \begin{bmatrix} 1 & \textbf{sgn}(x'_1) \\\\ \textbf{sgn}(x'_1) & 1 \end{bmatrix}$.
> > >
> > > Case $\epsilon >> \rho^{-1} :$, for $\epsilon >> \sigma >> \rho^{-1} $ we have  $H_{E, \sigma} \approx \begin{bmatrix} A & -\textbf{sgn}(x'_1) A \\\\ -\textbf{sgn} (x'_1) A & A \end{bmatrix}$ which yields $L^* = \begin{bmatrix} 1 & \textbf{sgn}(x'_1) \\\\ \textbf{sgn}(x'_1) & 1 \end{bmatrix}$.
> > >
> > > For $\sigma >> \epsilon$, we have $H_{E, \sigma} \approx \sigma^2 \rho^2 \begin{bmatrix} A & -\textbf{sgn}(x'_1) A \\\\ -\textbf{sgn} (x'_1) A & A \end{bmatrix} + \begin{bmatrix} 0 & 0 \\\\ 0 & C \end{bmatrix} $. But the second part is much smaller than the first and can be mostly ignored. Thus, we still get $L^* = \begin{bmatrix} 1 & \textbf{sgn}(x'_1) \\\\ \textbf{sgn}(x'_1) & 1 \end{bmatrix}$.
> > >
> > > Thus, we can choose the scale at which the symmetry is to be discovered.

---

### Official Review · Reviewer_9HNJ · 2025-03-13

**Overall Recommendation:** 4

**Summary:**

This paper introduces a data-free scheme for discovering the effective degrees of freedom in a molecular simulation, which is based upon the exploration of energy landscape symmetry. The effectiveness of this framework is theoretically illustrated and experimentally validated.

**Claims And Evidence:**

Yes. The theoretical and experimental arguments and evidence are very convincing. I especially like the idea of using just the energy, rather than data, to learn the symmetry.

**Essential References Not Discussed:**

NA.

**Experimental Designs Or Analyses:**

Yes. They look reasonable. Although figures 3-5 are somewhat too qualitative and can benefit from more quantitative arguments to be more closely related to the theoretical arguments.

**Methods And Evaluation Criteria:**

Yes. Although more thorough benchmarks using even smaller systems, such as Lenoard-Jones particles, charged particles, can be also interesting.

**Other Comments Or Suggestions:**

NA

**Other Strengths And Weaknesses:**

NA

**Questions For Authors:**

NA

**Relation To Broader Scientific Literature:**

The contributions are extremely relevant to the molecular simulation and drug discovery community.

**Theoretical Claims:**

Yes. They appear correct and sound.

---

> ### Author Rebuttal · Authors · 2025-04-01
>
> We thank the reviewer for their comments, and we try to address some of the concerns below:
>
> ### More Quantitative Results ###
> As pointed out by the reviewer, some of the results presented in the paper might look a bit qualitative.  Here, we provide some more quantitative statements about the results shown in the paper. We intend to add this table in the main paper in order to better illustrate the effectiveness of the discovered DOF.
>
> Long Simulation Results: The table below shows the frequency of observing the molecule in a specific conformation as a function of the simulation time ( simulation is run with a step size of 2 femtoseconds). We consider a molecule within a confirmation basin if the dihedral angles are within the known range for the specific conformer.
>
> |                              |          | 0.25 ns | 0.5 ns | 2.5 ns | 5 ns  | 25 ns | 50 ns | 250 ns | 500 ns |
> | ---------------------------- | -------- | ------- | ------ | ------ | ----- | ----- | ----- | ------ | ------ |
> |   | $C5$    | 0.257   | 0.268  | 0.233  | 0.247 | 0.246 | 0.238 | 0.215  | 0.23   |
> | Alanine Dipeptide in Vacuum |$C7_{eq}$                    | 0.479    | 0.429   | 0.448  | 0.436  | 0.436 | 0.425 | 0.382 | 0.408  |
> |  |$C7_{ax}$                    | 0        | 0       | 0      | 0      | 0     | 0.027 | 0.117 | 0.059  |
> |                              |          |         |        |        |       |       |       |        |        |
> | | $\\beta$ | 0.411   | 0.424  | 0.416  | 0.404 | 0.367 | 0.39  | 0.39   | 0.39   |
> | Alanine Dipeptide in Solvent  |$\\alpha_L$                  | 0        | 0       | 0      | 0      | 0.001 | 0.001 | 0.001 | 0.001  |
> | |$\\alpha_R$                  | 0.043    | 0.043   | 0.048  | 0.047  | 0.041 | 0.043 | 0.043 | 0.043  |
> |                              |          |         |        |        |       |       |       |        |        |
> | Chignolin in Solvent | folded   | 0.992   | 0.994  | 0.997  | 0.79  | 0.206 | 0.4   | 0.4    | 0.4    |
> | |misfolded                    | 0        | 0       | 0      | 0.183  | 0.073 | 0.048 | 0.048 | 0.048  |
>
> DOF-assisted Simulation Results: Under our method, we only run a simulation of 2 picosecond duration for every starting configuration provided by our DOF discovery algorithm. Thus, the conformers do not have enough time to escape from one conformational basin to another. Thus, the initial distribution of the conformers stays the same throughout the short simulation. Here, we provide the frequency of finding a molecule within a given conformational basin (within 20 ps of simulation) given the initial 1000 starting configurations.
>
> |                             |       | Full Hessian | Slow Hessian | Degenerate Hessian | Optimization eps 0.1_0.01 |
> | --------------------------- | ----- | ------- | -------- | --------- | ------------ |
> |  | $C5$ | 0.175   | 0.08     | 0.302     | 0.044        |
> | Alanine Dipeptide in Vacuum | $C7_{eq}$                   | 0.181 | 0.194   | 0.171    | 0.268     |
> |  |$C7_{ax}$                   | 0.214 | 0.194   | 0.134    | 0.231     |
>
> |                              |          | Full Hessian  | Slow Hessian | Degenerate Hessian | Optimization eps 0.1_0.01 | Optimization with solvent eps 0.1_0.01 |
> | ---------------------------- | -------- | ------- | -------- | --------- | ------------ | -------------------- |
> |  | $\\beta$ | 0.427   | 0.149    | 0.485     | 0.299        | 0.369                |
> | Alanine Dipeptide in Solvent | $\\alpha_L$                  | 0.069    | 0.144   | 0.022    | 0.207     | 0.157        |
> |  | $\\alpha_R$                  | 0.106    | 0.284   | 0.105    | 0.116     | 0.105        |
>
> |                      |        | Full Hessian  | Optimization eps 0.1_0.01 | Optimization eps 0.5_0.01 | Optimization with solvent eps 0.1_0.01 | Optimization with solvent eps 0.5_0.01 |
> | -------------------- | ------ | ------- | ------------ | ------------ | -------------------- | -------------------- |
> | Chignolin in Solvent | folded | 0.648   | 0.6          | 0.646        | 0.678                | 0.674                |
> | | misfolded            | 0.006  | 0.024   | 0.013        | 0.024        | 0.019                |
>
> ### Simulating the algorithm on additional smaller systems ###
> As suggested by the reviewer, we intend to add some example applications of the algorithm for smaller analytical systems. Please check bullet point 2 in our response to Reviewer p5r2 for further details and a small example system.

---

### Official Review · Reviewer_p5r2 · 2025-03-14

**Overall Recommendation:** 1

**Summary:**

This work proposes an approach for discovering degrees of freedom in MD simulations, relying primarily on the Hessian matrix rather than large simulation data. The method was applied to two prototypical peptide systems, showing that it is capable of efficient exploration of the configurations space along discovered DOFs.

**Claims And Evidence:**

The main results are on the qualitatively side. At least I am not sure how to interpret the plots in a quantitative way.

There is a lack of quantitative metrics. This is perhaps understandable given that the purpose is to discovery collective variables, a somewhat ambiguous goal.

**Essential References Not Discussed:**

Key literature survey of Hessian or approximate Hessian based methods for materials or molecular modeling is missing. Hessian based methods are extensively used in structure optimization and transition state finding/optimization. The latter is especially relevant for this paper.

**Experimental Designs Or Analyses:**

See above in "Claims And Evidence"

**Methods And Evaluation Criteria:**

There is no comparison with competing existing approaches.

See also above.

**Other Comments Or Suggestions:**

* About equation 4: The "direct optimization" uses Eq. 4, but it was stated that eq. 5 should be used instead of eq. 4.
* In "Full Hessian", Eq. 11 contains both sigma^2 and sigma^4. Was the whole of Eq. 11 used, or just the sigma^4 term?
* typo "a stable conformations"

**Other Strengths And Weaknesses:**

Strength: the method is geared towards an important scientific question of collective variable identification and conformation exploration for molecular system, with potential applications in drug design, protein folding dynamics, etc. An advantage of the method is that it does not require big MD simulation data.

Weakness: the presentation of the methodology is not easy to follow. Honestly I did not quite finish it. The the different methods to find the CVs were given without immediate explanation. There were some discussions/motivations scattered at other places. But it was not clear how to interpret the results properly without clear understanding of these methods. The results are lacking quantitative metrics and comparison with existing methods. This paper sounds like it has a lot of potential. But the present presentation prevents me from fully appreciating it.

**Questions For Authors:**

An easier to follow presentation of the theory could be very beneficial. A lot of jargon was invoked without explanation, which does not work for everybody.

**Relation To Broader Scientific Literature:**

There was literature review of several aspects. But the paper needs more discussions about existing Hessian based methods. See below

**Theoretical Claims:**

I tried to go through the derivations but got lost half way. So this part of my review should be considered incomplete.

---

> ### Author Rebuttal · Authors · 2025-04-01
>
> ### Literature Review for Hessian-based methods ###
> We thank the reviewer for pointing us toward this literature. We will include a detailed overview of the work in the literature and a comparison with the current paper. For geometric structure optimization, most of the methods in the literature build upon the idea of the Newton-Raphson method to accelerate gradient descent using second-order Hessian information to provide quadratic convergence near local minima by providing faster updates without calculating the full Hessian. On the other hand, transition state-finding methods utilize information about the eigenvector corresponding to a negative eigenvalue of the Hessian to find saddle points. Some synchronous transit methods like LST and QST also use Hessian information to better estimate the energy landscape near intermediate states between two final reaction states. While conformer discovery shares some of the same objectives as geometric structure optimization and transition state finding, unlike geometric structure optimization, we need to escape local energy basins to find a diverse set of conformers. Unlike the transition state finding task, we do not have a specified final reaction state that we intend to reach. Our primary objective is to discover certain local degrees of freedom to explore the energy landscape and find more conformers efficiently. In our method, we discover local DoF for a given conformer and use the discovered DoF to explore the basin near the conformer without any additional knowledge of CVs.
>
> ### Better Presentation of the Methodology ###
> As suggested by the reviewer, some of the methodology is quite hard to understand from the current presentation. We plan to reduce some of the jargon used in the paper and add a table of references in supplementary materials to help with the notation used in the paper. Additionally, we will add a section with some illustrative examples of applying our proposed methods to small systems. We present the following 1-d, 2-particle system suggested by Reviewer zCrM as an example. Due to space constraints, we only show the discovered local DoF using the Full Hessian method. We leave a more detailed discussion for larger systems in the paper. Consider the energy function: $E(x) = F(|x_1| - x_2) + \epsilon \sin\left(\frac{x_2}{\epsilon}\right)$ for $x \in \mathbb{R}^2$, where $F$ is a 1-D potential. In this case, we can use the first order optimality condition to see that at any minima, $F(|x_1| - x_2) = 0$ and $\cos\left(\frac{x_2}{\epsilon}\right) = 0$. Then, we see that the Hessian at any such minima, $x'$, can be given as
>
> $$ H_E(x') =  H_F(|x'_1| - x'_2) \begin{bmatrix} 1 & -\textbf{sgn}(x'_1) \\\\ -\textbf{sgn}(x'_1) &1 \end{bmatrix}
>     -  \frac{\cos(x'_2/\epsilon)}{\epsilon}\begin{bmatrix} 0 & 0 \\\\ 0 & 1\end{bmatrix} = C \begin{bmatrix} 1 & -\textbf{sgn}(x'_1) \\\\ -\textbf{sgn}(x'_1) & \gamma \end{bmatrix} $$
> for some constants $C, \gamma$.
> In order to find a local symmetry transformation $L$ such that for any $x$ near $x'$, we need $E(e^{\eta L} x) - E(x) \approx 0, \implies (\nabla E (x) \^\top L x)^2 \approx 0,\implies ((x-x')^\top H_E(x') L x)^2 \approx 0$.
>
> Full Hessian method needs us to minimize $2tr(K_S^2) + tr(K_S)^2$ where $K_S = (L^\top H_E (x') + H_E(x') L)/2$. This results in a generally complicated equation, but we get a simple solution for $\gamma \approx 1$. For $L^* = \frac{1}{2} \begin{bmatrix} 1 & \textbf{sgn}(x'_1)  \\\\ \textbf{sgn}(x'_1) &1 \end{bmatrix}$ gives $K_S = a\begin{bmatrix} 0 & -\textbf{sgn}(x'_1)\frac{1 - \gamma}{2} \\\\ -\textbf{sgn}(x'_1)\frac{1 - \gamma}{2} & 1-\gamma \end{bmatrix}$. Thus, $2tr(K_S^2) + tr(K_S)^2 = 4(1 - \gamma)^2 \approx 0$ for $\gamma \approx 1$. Thus, $L^*$ forms a local DoF for points around a minima $x'$.
>
> As ${L^*}^2 = L^*$, exponentiating this matrix we get, $e^{\eta L^*} = \sum_{j=0}^\infty \frac{\eta^j}{j!}{L^*}^j  = I + (\sum_{j=1}^\infty \frac{\eta^j}{j!})L^* = I + (e^{\eta} - 1)L^*$.
> Thus, we have $ e^{\eta L^*} x = x + (e^\eta - 1) L^* x = \begin{bmatrix} x_1 + (e^\eta - 1)(x_1 +  \textbf{sgn}(x'_1) x_2) & x_2 + (e^\eta - 1)(x_2 + \textbf{sgn}(x'_1) x_1) \end{bmatrix}^\top$. So, we see that if $x'_1 > 0, x_1 - x_2 = (e^{\eta L^*} x)_1 - (e^{\eta L^*} x)_2$ and if $x'_1 < 0, x_1 + x_2 = (e^{\eta L^*} x)_1 + (e^{\eta L^*} x)_2$ giving two different local DoF (conserved $x_1 - x_2$ when $x'_1 >0$ and conserved $x_1 + x_2$ when $x'_1 < 0 $) depending on the local structure of the hessian.
>
> ### Quantitative Results and Baselines ###
> Please refer to our response to Reviewer 9HNJ for additional results. Although there are existing methods for CV discovery, our method considers the slightly orthogonal task of discovering local DoF. Unlike CVs the discovered DoF are local in nature and do not necessarily correspond to any global Collective Variable. Thus, the proposed method is not directly comparable to existing baselines.

---

### Decision · Program_Chairs · 2025-05-01

**Decision:**

Accept (poster)

**Comment:**

The authors propose a new method to discover automatically degrees of freedom (DOF) in molecular dynamics simulations. The method does not rely on data, but seeks to compute instead approximate symmetries of the energy landscape. The method was praised for its novelty by all reviewers. Both reviewers that gave it the lowest score of "1 - Reject" still maintain that `This paper sounds like it has a lot of potential. But the present presentation prevents me from fully appreciating it.` or `I see potential in this paper`.

While I agree that the presentation is perfectible, I prefer to see the upsides in presenting a computationally novel approach. While Reviewer 9HNJ wrote something like a blank check (a strong 4 accept rating with little substance, and no follow-up post-rebuttal), I have taken into consideration the long review by Reviewer zCrM and their positive overall assessment of the paper. I see that the authors have taken steps to assuage some of the concerns expressed by Reviewer YyW7 on the presentation of experimental results.

For these reasons, I trust that the authors will take significant care in implementing **all** of the suggested presentation improvements, and I am overall supportive of this paper.